# GRADIENT DESCENT PROVABLY OPTIMIZES OVER-PARAMETERIZED NEURAL NETWORKS

**Simon S. Du**[*]
Machine Learning Department
Carnegie Mellon University
ssdu@cs.cmu.edu

**Xiyu Zhai**[*]
Department of EECS
Massachusetts Institute of Technology
xiyuzhai@mit.edu

**Barnabás Poczós**
Machine Learning Department
Carnegie Mellon University
bapozos@cs.cmu.edu

**Aarti Singh**
Machine Learning Department
Carnegie Mellon University
aartisingh@cmu.edu

## ABSTRACT

One of the mysteries in the success of neural networks is randomly initialized first order methods like gradient descent can achieve zero training loss even though the objective function is non-convex and non-smooth. This paper demystifies this surprising phenomenon for two-layer fully connected ReLU activated neural networks. For an $m$ hidden node shallow neural network with ReLU activation and $n$ training data, we show as long as $m$ is large enough and no two inputs are parallel, randomly initialized gradient descent converges to a *globally* optimal solution at a *linear* convergence rate for the quadratic loss function.

Our analysis relies on the following observation: over-parameterization and random initialization jointly restrict every weight vector to be close to its initialization for all iterations, which allows us to exploit a strong convexity-like property to show that gradient descent converges at a global linear rate to the global optimum. We believe these insights are also useful in analyzing deep models and other first order methods.

## 1 INTRODUCTION

Neural networks trained by first order methods have achieved a remarkable impact on many applications, but their theoretical properties are still mysteries. One of the empirical observation is even though the optimization objective function is non-convex and non-smooth, randomly initialized first order methods like stochastic gradient descent can still find a global minimum. Surprisingly, this property is not correlated with labels. In Zhang et al. (2016), authors replaced the true labels with randomly generated labels, but still found randomly initialized first order methods can always achieve zero training loss.

A widely believed explanation on why a neural network can fit all training labels is that the neural network is over-parameterized. For example, Wide ResNet (Zagoruyko and Komodakis) uses 100x parameters than the number of training data. Thus there must exist one such neural network of this architecture that can fit all training data. However, the existence does not imply why the network found by a randomly initialized first order method can fit all the data. The objective function is neither smooth nor convex, which makes traditional analysis technique from convex optimization not useful in this setting. To our knowledge, only the convergence to a stationary point is known (Davis et al., 2018).

---

[*]Equal contribution.

In this paper we demystify this surprising phenomenon on two-layer neural networks with rectified linear unit (ReLU) activation. Formally, we consider a neural network of the following form.

$$f(\mathbf{W}, \mathbf{a}, \mathbf{x}) = \frac{1}{\sqrt{m}} \sum_{r=1}^{m} a_r \sigma \left( \mathbf{w}_r^\top \mathbf{x} \right) \tag{1}$$

where $\mathbf{x} \in \mathbb{R}^d$ is the input, $\mathbf{w}_r \in \mathbb{R}^d$ is the weight vector of the first layer, $a_r \in \mathbb{R}$ is the output weight and $\sigma \left( \cdot \right)$ is the ReLU activation function: $\sigma \left( z \right) = z$ if $z \geq 0$ and $\sigma \left( z \right) = 0$ if $z < 0$.

We focus on the empirical risk minimization problem with a quadratic loss. Given a training data set $\{(\mathbf{x}_i, y_i)\}_{i=1}^n$, we want to minimize

$$L(\mathbf{W}, \mathbf{a}) = \sum_{i=1}^{n} \frac{1}{2} \left( f(\mathbf{W}, \mathbf{a}, \mathbf{x}_i) - y_i \right)^2. \tag{2}$$

Our main focus of this paper is to analyze the following procedure. We fix the second layer and apply gradient descent (GD) to optimize the first layer[1]

$$\mathbf{W}(k+1) = \mathbf{W}(k) - \eta \frac{\partial L(\mathbf{W}(k), \mathbf{a})}{\partial \mathbf{W}(k)}. \tag{3}$$

where $\eta > 0$ is the step size. Here the gradient formula for each weight vector is [2]

$$\frac{\partial L(\mathbf{W}, \mathbf{a})}{\partial \mathbf{w}_r} = \frac{1}{\sqrt{m}} \sum_{i=1}^{n} (f(\mathbf{W}, \mathbf{a}, \mathbf{x}_i) - y_i) \mathbf{a}_r \mathbf{x}_i \mathbb{I} \left\{ \mathbf{w}_r^\top \mathbf{x}_i \geq 0 \right\}. \tag{4}$$

Though this is only a shallow fully connected neural network, the objective function is still non-smooth and non-convex due to the use of ReLU activation function. [3] Even for this simple function, why randomly initialized first order method can achieve zero training error is not known. Many previous works have tried to answer this question or similar ones. Attempts include landscape analysis (Soudry and Carmon, 2016), partial differential equations (Mei et al.), analysis of the dynamics of the algorithm (Li and Yuan, 2017), optimal transport theory (Chizat and Bach, 2018), to name a few. These results often make strong assumptions on the labels and input distributions or do not imply why randomly initialized first order method can achieve zero training loss. See Section 2 for detailed comparisons between our result and previous ones.

In this paper, we rigorously prove that as long as no two inputs are parallel and $m$ is large enough, with randomly initialized $\mathbf{a}$ and $\mathbf{W}(0)$, gradient descent achieves zero training loss at a linear convergence rate, i.e., it finds a solution $\mathbf{W}(K)$ with $L(\mathbf{W}(K)) \leq \epsilon$ in $K = O(\log(1/\epsilon))$ iterations.[4] Thus, our theoretical result not only shows the global convergence but also gives a quantitative convergence rate in terms of the desired accuracy.

**Analysis Technique Overview** Our proof relies on the following insights. First we directly analyze the dynamics of each individual prediction $f(\mathbf{W}, \mathbf{a}, \mathbf{x}_i)$ for $i = 1, \ldots, n$. This is different from many previous work (Du et al., 2017b; Li and Yuan, 2017) which tried to analyze the dynamics of the parameter ($\mathbf{W}$) we are optimizing. Note because the objective function is non-smooth and non-convex, analysis of the parameter space dynamics is very difficult. In contrast, we find the dynamics of prediction space is governed by the spectral property of a Gram matrix (which can vary in each iteration, c.f. Equation (6)) and as long as this Gram matrix's least eigenvalue is lower bounded, gradient descent enjoys a linear rate. It is easy to show as long as no two inputs are parallel, in the initialization phase, this Gram matrix has a lower bounded least eigenvalue. (c.f. Theorem 3.1). Thus the problem reduces to showing the Gram matrix at later iterations is close to that in

---

[1]In Section 3.2, we also extend our technique to analyze the setting where we train both layers jointly.

[2] Note ReLU is not continuously differentiable. One can view $\frac{\partial L(\mathbf{W})}{\partial \mathbf{w}_r}$ as a convenient notation for the right hand side of (4) and this is the update rule used in practice.

[3]We remark that if one fixes the first layer and only optimizes the output layer, then the problem becomes a convex and smooth one. If $m$ is large enough, one can show the global minimum has zero training loss (Nguyen and Hein, 2018). Though for both cases (fixing the first layer and fixing the output layer), gradient descent achieves zero training loss, the learned prediction functions are different.

[4]Here we omit the polynomial dependency on $n$ and other data-dependent quantities.

the initialization phase. Our second observation is this Gram matrix is only related to the activation patterns ($\mathbb{I}\left\{\mathbf{w}_r^\top \mathbf{x}_i \geq 0\right\}$) and we can use matrix perturbation analysis to show if most of the patterns do not change, then this Gram matrix is close to its initialization. Our third observation is we find over-parameterization, random initialization, and the linear convergence jointly restrict every weight vector $\mathbf{w}_r$ to be close to its initialization. Then we can use this property to show most of the patterns do not change. Combining these insights we prove the first global quantitative convergence result of gradient descent on ReLU activated neural networks for the empirical risk minimization problem. Notably, our proof only uses linear algebra and standard probability bounds so we believe it can be easily generalized to analyze deep neural networks.

**Notations** We let $[n] = \{1, 2, \ldots, n\}$. Given a set $S$, we use $\text{unif}\left\{S\right\}$ to denote the uniform distribution over $S$. Given an event $E$, we use $\mathbb{I}\{A\}$ to be the indicator on whether this event happens. We use $N(\mathbf{0}, \mathbf{I})$ to denote the standard Gaussian distribution. For a matrix $\mathbf{A}$, we use $\mathbf{A}_{ij}$ to denote its $(i, j)$-th entry. We use $\|\cdot\|_2$ to denote the Euclidean norm of a vector, and use $\|\cdot\|_F$ to denote the Frobenius norm of a matrix. If a matrix $\mathbf{A}$ is positive semi-definite, we use $\lambda_{\min}(\mathbf{A})$ to denote its smallest eigenvalue. We use $\langle \cdot, \cdot \rangle$ to denote the standard Euclidean inner product between two vectors.

## 2 COMPARISON WITH PREVIOUS RESULTS

In this section, we survey an incomplete list of previous attempts in analyzing why first order methods can find a global minimum.

**Landscape Analysis** A popular way to analyze non-convex optimization problems is to identify whether the optimization landscape has some good geometric properties. Recently, researchers found if the objective function is smooth and satisfies (1) all local minima are global and (2) for every saddle point, there exists a negative curvature, then the noise-injected (stochastic) gradient descent (Jin et al., 2017; Ge et al., 2015; Du et al., 2017a) can find a global minimum in polynomial time. This algorithmic finding encouraged researchers to study whether the deep neural networks also admit these properties.

For the objective function defined in Equation (2), some partial results were obtained. Soudry and Carmon (2016) showed if $md \geq n$, then at every differentiable local minimum, the training error is zero. However, since the objective is non-smooth, it is hard to show gradient descent convergences to a differentiable local minimum. Xie et al. (2017) studied the same problem and related the loss to the gradient norm through the least singular value of the "extended feature matrix" $\mathbf{D}$ at the stationary points. However, they did not prove the convergence rate of the gradient norm. Interestingly, our analysis relies on the Gram matrix which is $\mathbf{D}\mathbf{D}^\top$.

Landscape analyses of ReLU activated neural networks for other settings have also been studied in many previous works (Ge et al., 2017; Safran and Shamir, 2016; Zhou and Liang, 2017; Freeman and Bruna, 2016; Hardt and Ma, 2016; Nguyen and Hein, 2018). These works establish favorable landscape properties but none of them implies that gradient descent converges to a global minimizer of the empirical risk. More recently, some negative results have also been discovered (Safran and Shamir, 2018; Yun et al., 2018a) and new procedures have been proposed to test local optimality and escape strict saddle points at non-differentiable points (Yun et al., 2018b). However, the new procedures cannot find global minima as well. For other activation functions, some previous works showed the landscape does have the desired geometric properties (Du and Lee, 2018; Soltanolkotabi et al., 2018; Nguyen and Hein, 2017; Kawaguchi, 2016; Haeffele and Vidal, 2015; Andoni et al., 2014; Venturi et al., 2018; Yun et al., 2018a). However, it is unclear how to extend their analyses to our setting.

**Analysis of Algorithm Dynamics** Another way to prove convergence result is to analyze the dynamics of first order methods directly. Our paper also belongs to this category. Many previous works assumed (1) the input distribution is Gaussian and (2) the label is generated according to a planted neural network. Based on these two (unrealistic) conditions, it can be shown that randomly initialized (stochastic) gradient descent can learn a ReLU (Tian, 2017; Soltanolkotabi, 2017), a single convolutional filter (Brutzkus and Globerson, 2017), a convolutional neural network with one filter and one output layer (Du et al., 2018b) and residual network with small spectral norm weight

matrix (Li and Yuan, 2017).[5] Beyond Gaussian input distribution, Du et al. (2017b) showed for learning a convolutional filter, the Gaussian input distribution assumption can be relaxed but they still required the label is generated from an underlying true filter. Comparing with these work, our paper does not try to recover the underlying true neural network. Instead, we focus on providing theoretical justification on why randomly initialized gradient descent can achieve zero training loss, which is what we can observe and verify in practice.

Jacot et al. (2018) established an asymptotic result showing for the multilayer fully-connected neural network with a smooth activation function, if every layer's weight matrix is infinitely wide, then for finite training time, the convergence of gradient descent can be characterized by a kernel. Our proof technique relies on a Gram matrix which is the kernel matrix in their paper. Our paper focuses on the two-layer neural network with ReLU activation function (non-smooth) and we are able to prove the Gram matrix is stable for infinite training time.

The most related paper is by Li and Liang (2018) who observed that when training a two-layer full connected neural network, most of the patterns ($\mathbb{I}\left\{\mathbf{w}_r^\top \mathbf{x}_i \geq 0\right\}$) do not change over iterations, which we also use to show the stability of the Gram matrix. They used this observation to obtain the convergence rate of GD on a two-layer over-parameterized neural network for the cross-entropy loss. They need the number of hidden nodes $m$ scales with $\text{poly}(1/\epsilon)$ where $\epsilon$ is the desired accuracy. Thus unless the number of hidden nodes $m \to \infty$, their result does not imply GD can achieve zero training loss. We improve by allowing the amount of over-parameterization to be independent of the desired accuracy and show GD can achieve zero training loss. Furthermore, our proof is much simpler and more transparent so we believe it can be easily generalized to analyze other neural network architectures.

**Other Analysis Approaches** Chizat and Bach (2018) used optimal transport theory to analyze continuous time gradient descent on over-parameterized models. They required the second layer to be infinitely wide and their results on ReLU activated neural network is only at the formal level. Mei et al. analyzed SGD for optimizing the population loss and showed the dynamics can be captured by a partial differential equation in the suitable scaling limit. They listed some specific examples on input distributions including mixture of Gaussians. However, it is still unclear whether this framework can explain why first order methods can minimize the empirical risk. Daniely (2017) built connection between neural networks with kernel methods and showed stochastic gradient descent can learn a function that is competitive with the best function in the conjugate kernel space of the network. Again this work does not imply why first order methods can achieve zero training loss.

## 3 CONTINUOUS TIME ANALYSIS

In this section, we present our result for gradient flow, i.e., gradient descent with infinitesimal step size. The analysis of gradient flow is a stepping stone towards understanding discrete algorithms and this is the main topic of recent work (Arora et al., 2018; Du et al., 2018a). In the next section, we will modify the proof and give a quantitative bound for gradient descent with positive step size. Formally, we consider the ordinary differential equation[6] defined by:

$$\frac{d\mathbf{w}_r(t)}{dt} = -\frac{\partial L(\mathbf{W}(t), \mathbf{a})}{\partial \mathbf{w}_r(t)}$$

for $r \in [m]$. We denote $u_i(t) = f(\mathbf{W}(t), \mathbf{a}, \mathbf{x}_i)$ the prediction on input $\mathbf{x}_i$ at time $t$ and we let $\mathbf{u}(t) = (u_1(t), \ldots, u_n(t)) \in \mathbb{R}^n$ be the prediction vector at time $t$. We state our main assumption.

**Assumption 3.1.** *Define matrix* $\mathbf{H}^\infty \in \mathbb{R}^{n \times n}$ *with* $\mathbf{H}_{ij}^\infty = \mathbb{E}_{\mathbf{w} \sim N(\mathbf{0}, \mathbf{I})}\left[\mathbf{x}_i^\top \mathbf{x}_j \mathbb{I}\left\{\mathbf{w}^\top \mathbf{x}_i \geq 0, \mathbf{w}^\top \mathbf{x}_j \geq 0\right\}\right]$. *We assume* $\lambda_0 \triangleq \lambda_{\min}\left(\mathbf{H}^\infty\right) > 0$.

$\mathbf{H}^\infty$ is the Gram matrix induced by the ReLU activation function and the random initialization. Later we will show that during the training, though the Gram matrix may change (c.f. Equation (6)), it is still close to $\mathbf{H}^\infty$. Furthermore, as will be apparent in the proof (c.f. Equation (7)), $\mathbf{H}^\infty$ is

---

[5]Since these work assume the label is realizable, converging to global minimum is equivalent to recovering the underlying model.

[6]Strictly speaking, this should be differential inclusion (Davis et al., 2018)

the fundamental quantity that determines the convergence rate. Interestingly, various properties of this $\mathbf{H}^\infty$ matrix has been studied in previous works (Xie et al., 2017; Tsuchida et al., 2017). Now to justify this assumption, the following theorem shows if no two inputs are parallel the least eigenvalue is strictly positive.

**Theorem 3.1.** *If for any $i \neq j$, $\mathbf{x}_i \not\parallel \mathbf{x}_j$, then $\lambda_0 > 0$.*

Note for most real world datasets, no two inputs are parallel, so our assumption holds in general. Now we are ready to state our main theorem in this section.

**Theorem 3.2** (Convergence Rate of Gradient Flow)**.** *Suppose Assumption 3.1 holds and for all $i \in [n]$, $\|\mathbf{x}_i\|_2 = 1$ and $|y_i| \leq C$ for some constant $C$. Then if we set the number of hidden nodes $m = \Omega\left(\frac{n^6}{\lambda_0^4 \delta^3}\right)$ and we i.i.d. initialize $\mathbf{w}_r \sim N(\mathbf{0}, \mathbf{I})$, $a_r \sim \text{unif}\left[\{-1, 1\}\right]$ for $r \in [m]$, then with probability at least $1 - \delta$ over the initialization, we have*

$$\|\mathbf{u}(t) - \mathbf{y}\|_2^2 \leq \exp(-\lambda_0 t) \|\mathbf{u}(0) - \mathbf{y}\|_2^2.$$

This theorem establishes that if $m$ is large enough, the training error converges to $0$ at a linear rate. Here we assume $\|\mathbf{x}_i\|_2 = 1$ only for simplicity and it is not hard to relax this condition.[7] The bounded label condition also holds for most real world data set. The number of hidden nodes $m$ required is $\Omega\left(\frac{n^6}{\lambda_0^4 \delta^3}\right)$, which depends on the number of samples $n$, $\lambda_0$, and the failure probability $\delta$. Over-parameterization, i.e., the fact $m = \text{poly}(n, 1/\lambda_0, 1/\delta)$, plays a crucial role in guaranteeing gradient descent to find the global minimum. In this paper, we only use the simplest concentration inequalities (Hoeffding's and Markov's) in order to have the cleanest proof. We believe using a more advanced concentration analysis we can further improve the dependency. Lastly, we note the specific convergence rate depends on $\lambda_0$ but independent of the number of hidden nodes $m$.

### 3.1 PROOF OF THEOREM 3.2

Our first step is to calculate the dynamics of each prediction.

$$\frac{d}{dt} u_i(t) = \sum_{r=1}^m \langle \frac{\partial f(\mathbf{W}(t), \mathbf{a}, \mathbf{x}_i)}{\partial \mathbf{w}_r(t)}, \frac{d\mathbf{w}_r(t)}{dt} \rangle$$

$$= \sum_{j=1}^n (y_j - u_j) \sum_{r=1}^m \langle \frac{\partial f(\mathbf{W}(t), \mathbf{a}, \mathbf{x}_i)}{\partial \mathbf{w}_r(t)}, \frac{\partial f(\mathbf{W}(t), \mathbf{a}, \mathbf{x}_j)}{\partial \mathbf{w}_r(t)} \rangle \triangleq \sum_{j=1}^n (y_j - u_j) \mathbf{H}_{ij}(t) \quad (5)$$

where $\mathbf{H}(t)$ is an $n \times n$ matrix with $(i, j)$-th entry

$$\mathbf{H}_{ij}(t) = \frac{1}{m} \mathbf{x}_i^\top \mathbf{x}_j \sum_{r=1}^m \mathbb{I}\left\{\mathbf{x}_i^\top \mathbf{w}_r(t) \geq 0, \mathbf{x}_j^\top \mathbf{w}_r(t) \geq 0\right\}. \quad (6)$$

With this $\mathbf{H}(t)$ matrix, we can write the dynamics of predictions in a compact way:

$$\frac{d}{dt} \mathbf{u}(t) = \mathbf{H}(t)(\mathbf{y} - \mathbf{u}(t)). \quad (7)$$

**Remark 3.1.** *Note Equation (7) completely describes the dynamics of the predictions. In the rest of this section, we will show (1) at initialization $\|\mathbf{H}(0) - \mathbf{H}^\infty\|_2$ is $O(\sqrt{1/m})$ and (2) for all $t > 0$, $\|\mathbf{H}(t) - \mathbf{H}(0)\|_2$ is $O(\sqrt{1/m})$. Therefore, according to Equation (7), as $m \to \infty$, the dynamics of the predictions are characterized by $\mathbf{H}^\infty$. This is the main reason we believe $\mathbf{H}^\infty$ is the fundamental quantity that describes this optimization process.*

$\mathbf{H}(t)$ is a time-dependent symmetric matrix. We first analyze its property when $t = 0$. The following lemma shows if $m$ is large then $\mathbf{H}(0)$ has a lower bounded least eigenvalue with high probability. The proof is by the standard concentration bound so we defer it to the appendix.

---

[7] More precisely, if $0 < c_{low} \leq \|\mathbf{x}_i\|_2 \leq c_{high}$ for all $i \in [n]$, we only need to change Lemma 3.1-3.3 to make them depend on $c_{low}$ and $c_{high}$ and the amount of over-parameterization $m$ will depend on $\frac{c_{high}}{c_{low}}$. We assume $\|\mathbf{x}_i\|_2 = 1$ so we can present the cleanest proof and focus on our main analysis technique.

**Lemma 3.1.** *If $m = \Omega\left(\frac{n^2}{\lambda_0^2}\log\left(\frac{n}{\delta}\right)\right)$, we have with probability at least $1-\delta$, $\|\mathbf{H}(0) - \mathbf{H}^\infty\|_2 \leq \frac{\lambda_0}{4}$ and $\lambda_{\min}(\mathbf{H}(0)) \geq \frac{3}{4}\lambda_0$.*

Our second step is to show $\mathbf{H}(t)$ is stable in terms of $\mathbf{W}(t)$. Formally, the following lemma shows for any $\mathbf{W}$ close to $\mathbf{W}(0)$, the induced Gram matrix $\mathbf{H}$ is close to $\mathbf{H}(0)$ and has a lower bounded least eigenvalue.

**Lemma 3.2.** *If $\mathbf{w}_1, \ldots, \mathbf{w}_m$ are i.i.d. generated from $N(\mathbf{0}, \mathbf{I})$, then with probability at least $1 - \delta$, the following holds. For any set of weight vectors $\mathbf{w}_1, \ldots, \mathbf{w}_m \in \mathbb{R}^d$ that satisfy for any $r \in [m]$, $\|\mathbf{w}_r(0) - \mathbf{w}_r\|_2 \leq \frac{c\delta\lambda_0}{n^2} \triangleq R$ for some small positive constant c, then the matrix $\mathbf{H} \in \mathbb{R}^{n \times n}$ defined by*

$$\mathbf{H}_{ij} = \frac{1}{m}\mathbf{x}_i^\top \mathbf{x}_j \sum_{r=1}^{m} \mathbb{I}\left\{\mathbf{w}_r^\top \mathbf{x}_i \geq 0, \mathbf{w}_r^\top \mathbf{x}_j \geq 0\right\}$$

*satisfies $\|\mathbf{H} - \mathbf{H}(0)\|_2 < \frac{\lambda_0}{4}$ and $\lambda_{\min}(\mathbf{H}) > \frac{\lambda_0}{2}$.*

This lemma plays a crucial role in our analysis so we give the proof below.

*Proof of Lemma 3.2* We define the event

$$A_{ir} = \left\{\exists \mathbf{w} : \|\mathbf{w} - \mathbf{w}_r(0)\| \leq R, \mathbb{I}\left\{\mathbf{x}_i^\top \mathbf{w}_r(0) \geq 0\right\} \neq \mathbb{I}\left\{\mathbf{x}_i^\top \mathbf{w} \geq 0\right\}\right\}.$$

Note this event happens if and only if $\left|\mathbf{w}_r(0)^\top \mathbf{x}_i\right| < R$. Recall $\mathbf{w}_r(0) \sim N(\mathbf{0}, \mathbf{I})$. By anti-concentration inequality of Gaussian, we have $P(A_{ir}) = P_{z \sim N(0,1)}\left(|z| < R\right) \leq \frac{2R}{\sqrt{2\pi}}$. Therefore, for any set of weight vectors $\mathbf{w}_1, \ldots, \mathbf{w}_m$ that satisfy the assumption in the lemma, we can bound the entry-wise deviation on their induced matrix $\mathbf{H}$: for any $(i, j) \in [n] \times [n]$

$$\mathbb{E}\left[\left|\mathbf{H}_{ij}(0) - \mathbf{H}_{ij}\right|\right]$$

$$= \mathbb{E}\left[\frac{1}{m}\left|\mathbf{x}_i^\top \mathbf{x}_j \sum_{r=1}^{m}\left(\mathbb{I}\left\{\mathbf{w}_r(0)^\top \mathbf{x}_i \geq 0, \mathbf{w}_r(0)^\top \mathbf{x}_j \geq 0\right\} - \mathbb{I}\left\{\mathbf{w}_r^\top \mathbf{x}_i \geq 0, \mathbf{w}_r^\top \mathbf{x}_j \geq 0\right\}\right)\right|\right]$$

$$\leq \frac{1}{m}\sum_{r=1}^{m}\mathbb{E}\left[\mathbb{I}\left\{A_{ir} \cup A_{jr}\right\}\right] \leq \frac{4R}{\sqrt{2\pi}}$$

where the expectation is taken over the random initialization of $\mathbf{w}_1(0), \ldots, \mathbf{w}_m(0)$. Summing over $(i, j)$, we have $\mathbb{E}\left[\sum_{(i,j)=(1,1)}^{(n,n)}\left|\mathbf{H}_{ij} - \mathbf{H}_{ij}(0)\right|\right] \leq \frac{4n^2 R}{\sqrt{2\pi}}$. Thus by Markov's inequality, with probability $1 - \delta$, we have $\sum_{(i,j)=(1,1)}^{(n,n)}\left|\mathbf{H}_{ij} - \mathbf{H}_{ij}(0)\right| \leq \frac{4n^2 R}{\sqrt{2\pi}\delta}$. Next, we use matrix perturbation theory to bound the deviation from the initialization

$$\|\mathbf{H} - \mathbf{H}(0)\|_2 \leq \|\mathbf{H} - \mathbf{H}(0)\|_F \leq \sum_{(i,j)=(1,1)}^{(n,n)}\left|\mathbf{H}_{ij} - \mathbf{H}_{ij}(0)\right| \leq \frac{4n^2 R}{\sqrt{2\pi}\delta}.$$

Lastly, we lower bound the smallest eigenvalue by plugging in $R$

$$\lambda_{\min}(\mathbf{H}) \geq \lambda_{\min}(\mathbf{H}(0)) - \frac{4n^2 R}{\sqrt{2\pi}\delta} \geq \frac{\lambda_0}{2}. \quad \square$$

The next lemma shows two facts if the least eigenvalue of $\mathbf{H}(t)$ is lower bounded. First, the loss converges to 0 at a linear convergence rate. Second, $\mathbf{w}_r(t)$ is close to the initialization for every $r \in [m]$. This lemma clearly demonstrates the power of over-parameterization.

**Lemma 3.3.** *Suppose for $0 \leq s \leq t$, $\lambda_{\min}(\mathbf{H}(s)) \geq \frac{\lambda_0}{2}$. Then we have $\|\mathbf{y} - \mathbf{u}(t)\|_2^2 \leq \exp(-\lambda_0 t)\|\mathbf{y} - \mathbf{u}(0)\|_2^2$ and for any $r \in [m]$, $\|\mathbf{w}_r(t) - \mathbf{w}_r(0)\|_2 \leq \frac{\sqrt{n}\|\mathbf{y}-\mathbf{u}(0)\|_2}{\sqrt{m}\lambda_0} \triangleq R'$.*

*Proof of Lemma 3.3* Recall we can write the dynamics of predictions as $\frac{d}{dt}\mathbf{u}(t) = \mathbf{H}(\mathbf{y} - \mathbf{u}(t))$. We can calculate the loss function dynamics

$$\frac{d}{dt}\|\mathbf{y} - \mathbf{u}(t)\|_2^2 = -2(\mathbf{y} - \mathbf{u}(t))^\top \mathbf{H}(t)(\mathbf{y} - \mathbf{u}(t))$$

$$\leq -\lambda_0 \|\mathbf{y} - \mathbf{u}(t)\|_2^2.$$

Thus we have $\frac{d}{dt}\left(\exp(\lambda_0 t)\|\mathbf{y}-\mathbf{u}(t)\|_2^2\right) \leq 0$ and $\exp(\lambda_0 t)\|\mathbf{y}-\mathbf{u}(t)\|_2^2$ is a decreasing function with respect to $t$. Using this fact we can bound the loss

$$\|\mathbf{y}-\mathbf{u}(t)\|_2^2 \leq \exp(-\lambda_0 t)\|\mathbf{y}-\mathbf{u}(0)\|_2^2.$$

Therefore, $\mathbf{u}(t) \to \mathbf{y}$ exponentially fast. Now we bound the gradient norm. Recall for $0 \leq s \leq t$,

$$\left\|\frac{d}{ds}\mathbf{w}_r(s)\right\|_2 = \left\|\sum_{i=1}^n (y_i - u_i)\frac{1}{\sqrt{m}}a_r\mathbf{x}_i\mathbb{I}\left\{\mathbf{w}_r(s)^\top\mathbf{x}_i \geq 0\right\}\right\|_2$$

$$\leq \frac{1}{\sqrt{m}}\sum_{i=1}^n |y_i - u_i(s)| \leq \frac{\sqrt{n}}{\sqrt{m}}\|\mathbf{y}-\mathbf{u}(s)\|_2 \leq \frac{\sqrt{n}}{\sqrt{m}}\exp(-\lambda_0 s)\|\mathbf{y}-\mathbf{u}(0)\|_2.$$

Integrating the gradient, we can bound the distance from the initialization

$$\|\mathbf{w}_r(t) - \mathbf{w}_r(0)\|_2 \leq \int_0^t \left\|\frac{d}{ds}\mathbf{w}_r(s)\right\|_2 ds \leq \frac{\sqrt{n}\|\mathbf{y}-\mathbf{u}(0)\|_2}{\sqrt{m}\lambda_0}. \quad \square$$

The next lemma shows if $R' < R$, the conditions in Lemma 3.2 and 3.3 hold for all $t \geq 0$. The proof is by contradiction and we defer it to appendix.

**Lemma 3.4.** *If $R' < R$, we have for all $t \geq 0$, $\lambda_{\min}(\mathbf{H}(t)) \geq \frac{1}{2}\lambda_0$, for all $r \in [m]$, $\|\mathbf{w}_r(t) - \mathbf{w}_r(0)\|_2 \leq R'$ and $\|\mathbf{y}-\mathbf{u}(t)\|_2^2 \leq \exp(-\lambda_0 t)\|\mathbf{y}-\mathbf{u}(0)\|_2^2$.*

Thus it is sufficient to show $R' < R$ which is equivalent to $m = \Omega\left(\frac{n^5\|\mathbf{y}-\mathbf{u}(0)\|_2^2}{\lambda_0^4\delta^2}\right)$. We bound

$$\mathbb{E}\left[\|\mathbf{y}-\mathbf{u}(0)\|_2^2\right] = \sum_{i=1}^n (y_i^2 + y_i\mathbb{E}\left[f(\mathbf{W}(0),\mathbf{a},\mathbf{x}_i)\right] + \mathbb{E}\left[f(\mathbf{W}(0),\mathbf{a},\mathbf{x}_i)^2\right]) = \sum_{i=1}^n (y_i^2 + 1) = O(n).$$

Thus by Markov's inequality, we have with probability at least $1-\delta$, $\|\mathbf{y}-\mathbf{u}(0)\|_2^2 = O(\frac{n}{\delta})$. Plugging in this bound we prove the theorem. $\quad\square$

## 3.2 JOINTLY TRAINING BOTH LAYERS

In this subsection, we showcase our proof technique can be applied to analyze the convergence of gradient flow for jointly training both layers. Formally, we consider the ordinary differential equation defined by:

$$\frac{d\mathbf{w}_r(t)}{dt} = -\frac{\partial L(\mathbf{W}(t),\mathbf{a}(t))}{\partial\mathbf{w}_r(t)} \text{ and } \frac{d\mathbf{w}_r(t)}{dt} = -\frac{\partial L(\mathbf{W}(t),\mathbf{a}(t))}{\partial\mathbf{a}_r(t)}$$

for $r = 1,\ldots,m$. The following theorem shows using gradient flow to jointly train both layers, we can still enjoy linear convergence rate towards zero loss.

**Theorem 3.3** (Convergence Rate of Gradient Flow for Training Both Layers)**.** *Under the same assumptions as in Theorem 3.2, if we set the number of hidden nodes $m = \Omega\left(\frac{n^6\log(m/\delta)}{\lambda_0^4\delta^3}\right)$ and we i.i.d. initialize $\mathbf{w}_r \sim N(\mathbf{0},\mathbf{I})$, $a_r \sim \text{unif}\left[\{-1,1\}\right]$ for $r \in [m]$, with probability at least $1-\delta$ over the initialization we have*

$$\|\mathbf{u}(t) - \mathbf{y}\|_2^2 \leq \exp(-\lambda_0 t)\|\mathbf{u}(0) - \mathbf{y}\|_2^2.$$

Theorem 3.3 shows under the same assumptions as in Theorem 3.2, we can achieve the same convergence rate as that of only training the first layer. The proof of Theorem 3.3 relies on the same arguments as the proof of Theorem 3.2. Again we consider the dynamics of the predictions and this dynamics is characterized by a Gram matrix. We can show for all $t > 0$, this Gram matrix is close to the Gram matrix at the initialization phase. We refer readers to appendix for the full proof.

## 4 DISCRETE TIME ANALYSIS

In this section, we show randomly initialized gradient descent with a constant positive step size converges to the global minimum at a linear rate. We first present our main theorem.

**Theorem 4.1** (Convergence Rate of Gradient Descent). *Under the same assumptions as in Theorem 3.2, if we set the number of hidden nodes $m = \Omega\left(\frac{n^6}{\lambda_0^4 \delta^3}\right)$, we i.i.d. initialize $\mathbf{w}_r \sim N(\mathbf{0}, \mathbf{I})$, $a_r \sim \mathrm{unif}\left[\{-1, 1\}\right]$ for $r \in [m]$, and we set the step size $\eta = O\left(\frac{\lambda_0}{n^2}\right)$ then with probability at least $1 - \delta$ over the random initialization we have for $k = 0, 1, 2, \ldots$*

$$\|\mathbf{u}(k) - \mathbf{y}\|_2^2 \le \left(1 - \frac{\eta\lambda_0}{2}\right)^k \|\mathbf{u}(0) - \mathbf{y}\|_2^2.$$

Theorem 4.1 shows even though the objective function is non-smooth and non-convex, gradient descent with a constant step size still enjoys a linear convergence rate. Our assumptions on the least eigenvalue and the number of hidden nodes are exactly the same as the theorem for gradient flow.

### 4.1 PROOF OF THEOREM 4.1

We prove Theorem 4.1 by induction. Our induction hypothesis is just the following convergence rate of the empirical loss.

**Condition 4.1.** *At the $k$-th iteration, we have $\|\mathbf{y} - \mathbf{u}(k)\|_2^2 \le (1 - \frac{\eta\lambda_0}{2})^k \|\mathbf{y} - \mathbf{u}(0)\|_2^2$.*

A directly corollary of this condition is the following bound of deviation from the initialization. The proof is similar to that of Lemma 3.3 so we defer it to appendix.

**Corollary 4.1.** *If Condition 4.1 holds for $k' = 0, \ldots, k$, then we have for every $r \in [m]$*

$$\|\mathbf{w}_r(k+1) - \mathbf{w}_r(0)\|_2 \le \frac{4\sqrt{n} \|\mathbf{y} - \mathbf{u}(0)\|_2}{\sqrt{m}\lambda_0} \triangleq R'. \tag{8}$$

Now we show Condition 4.1 holds for every $k = 0, 1, \ldots$. For the base case $k = 0$, by definition Condition 4.1 holds. Suppose for $k' = 0, \ldots, k$, Condition 4.1 holds and we want to show Condition 4.1 holds for $k' = k + 1$.

Our strategy is similar to the proof of Theorem 3.2. We define the event

$$A_{ir} = \left\{ \exists \mathbf{w} : \|\mathbf{w} - \mathbf{w}_r(0)\| \le R, \mathbb{I}\left\{\mathbf{x}_i^\top \mathbf{w}_r(0) \ge 0\right\} \ne \mathbb{I}\left\{\mathbf{x}_i^\top \mathbf{w} \ge 0\right\} \right\}.$$

where $R = \frac{c\lambda_0}{n^2}$ for some small positive constant $c$. Different from gradient flow, for gradient descent we need a more refined analysis. We let $S_i = \{r \in [m] : \mathbb{I}\{A_{ir}\} = 0\}$ and $S_i^\perp = [m] \setminus S_i$. The following lemma bounds the sum of sizes of $S_i^\perp$. The proof is similar to the analysis used in Lemma 3.2. See Section A for the whole proof.

**Lemma 4.1.** *With probability at least $1 - \delta$ over the initialization, we have $\sum_{i=1}^n \left|S_i^\perp\right| \le \frac{CmnR}{\delta}$ for some positive constant $C > 0$.*

Next, we calculate the difference of predictions between two consecutive iterations, analogue to $\frac{du_i(t)}{dt}$ term in Section 3.

$$u_i(k+1) - u_i(k) = \frac{1}{\sqrt{m}} \sum_{r=1}^m a_r \left(\sigma\left(\mathbf{w}_r(k+1)^\top \mathbf{x}_i\right) - \sigma\left(\mathbf{w}_r(k)^\top \mathbf{x}_i\right)\right)$$

$$= \frac{1}{\sqrt{m}} \sum_{r=1}^m a_r \left(\sigma\left(\left(\mathbf{w}_r(k) - \eta\frac{\partial L(\mathbf{W}(k))}{\partial \mathbf{w}_r(k)}\right)^\top \mathbf{x}_i\right) - \sigma\left(\mathbf{w}_r(k)^\top \mathbf{x}_i\right)\right).$$

Here we divide the right hand side into two parts. $I_1^i$ accounts for terms that the pattern does not change and $I_2^i$ accounts for terms that pattern may change.

$$I_1^i \triangleq \frac{1}{\sqrt{m}} \sum_{r \in S_i} a_r \left( \sigma \left( \left( \mathbf{w}_r(k) - \eta \frac{\partial L(\mathbf{W}(k))}{\partial \mathbf{w}_r(k)} \right)^\top \mathbf{x}_i \right) - \sigma \left( \mathbf{w}_r(k)^\top \mathbf{x}_i \right) \right)$$

$$I_2^i \triangleq \frac{1}{\sqrt{m}} \sum_{r \in S_i^\perp} a_r \left( \sigma \left( \left( \mathbf{w}_r(k) - \eta \frac{\partial L(\mathbf{W}(k))}{\partial \mathbf{w}_r(k)} \right)^\top \mathbf{x}_i \right) - \sigma \left( \mathbf{w}_r(k)^\top \mathbf{x}_i \right) \right)$$

We view $I_2^i$ as a perturbation and bound its magnitude. Because ReLU is a 1-Lipschitz function and $|a_r| = 1$, we have

$$|I_2^i| \leq \frac{\eta}{\sqrt{m}} \sum_{r \in S_i^\perp} \left| \left( \frac{\partial L(\mathbf{W}(k))}{\partial \mathbf{w}_r(k)} \right)^\top \mathbf{x}_i \right| \leq \frac{\eta |S_i^\perp|}{\sqrt{m}} \max_{r \in [m]} \left\| \frac{\partial L(\mathbf{W}(k))}{\partial \mathbf{w}_r(k)} \right\|_2 \leq \frac{\eta |S_i|^\perp \sqrt{n} \|\mathbf{u}(k) - \mathbf{y}\|_2}{m}.$$

To analyze $I_1^i$, by Corollary 4.1, we know $\|\mathbf{w}_r(k) - \mathbf{w}_r(0)\| \leq R'$ and $\|\mathbf{w}_r(k) - \mathbf{w}_r(0)\| \leq R'$ for all $r \in [m]$. Furthermore, because $R' < R$, we know $\mathbb{I}\left\{ \mathbf{w}_r(k+1)^\top \mathbf{x}_i \geq 0 \right\} = \mathbb{I}\left\{ \mathbf{w}_r(k)^\top \mathbf{x}_i \geq 0 \right\}$ for $r \in S_i$. Thus we can find a more convenient expression of $I_1^i$ for analysis

$$I_1^i = -\frac{\eta}{m} \sum_{j=1}^n \mathbf{x}_i^\top \mathbf{x}_j (u_j - y_j) \sum_{r \in S_i} \mathbb{I}\left\{ \mathbf{w}_r(k)^\top \mathbf{x}_i \geq 0, \mathbf{w}_r(k)^\top \mathbf{x}_j \geq 0 \right\}$$

$$= -\eta \sum_{j=1}^n (u_j - y_j)(\mathbf{H}_{ij}(k) - \mathbf{H}_{ij}^\perp(k))$$

where $\mathbf{H}_{ij}(k) = \frac{1}{m} \sum_{r=1}^m \mathbf{x}_i^\top \mathbf{x}_j \mathbb{I}\left\{ \mathbf{w}_r(k)^\top \mathbf{x}_i \geq 0, \mathbf{w}_r(k)^\top \mathbf{x}_j \geq 0 \right\}$ is just the $(i,j)$-th entry of a discrete version of Gram matrix defined in Section 3 and $\mathbf{H}_{ij}^\perp(k) = \frac{1}{m} \sum_{r \in S_i^\perp} \mathbf{x}_i^\top \mathbf{x}_j \mathbb{I}\left\{ \mathbf{w}_r(k)^\top \mathbf{x}_i \geq 0, \mathbf{w}_r(k)^\top \mathbf{x}_j \geq 0 \right\}$ is a perturbation matrix. Let $\mathbf{H}^\perp(k)$ be the $n \times n$ matrix with $(i,j)$-th entry being $\mathbf{H}_{ij}^\perp(k)$. Using Lemma 4.1, we obtain an upper bound of the operator norm

$$\left\| \mathbf{H}^\perp(k) \right\|_2 \leq \sum_{(i,j)=(1,1)}^{(n,n)} \left| \mathbf{H}_{ij}^\perp(k) \right| \leq \frac{n \sum_{i=1}^n |S_i^\perp|}{m} \leq \frac{Cn^2 mR}{\delta m} \leq \frac{Cn^2 R}{\delta}.$$

Similar to the classical analysis of gradient descent, we also need bound the quadratic term.

$$\|\mathbf{u}(k+1) - \mathbf{u}(k)\|_2^2 \leq \eta^2 \sum_{i=1}^n \frac{1}{m} \left( \sum_{r=1}^m \left\| \frac{\partial L(\mathbf{W}(k))}{\partial \mathbf{w}_r(k)} \right\|_2 \right)^2 \leq \eta^2 n^2 \|\mathbf{y} - \mathbf{u}(k)\|_2^2.$$

With these estimates at hand, we are ready to prove the induction hypothesis.

$$\begin{aligned}
\|\mathbf{y} - \mathbf{u}(k+1)\|_2^2 &= \|\mathbf{y} - \mathbf{u}(k) - (\mathbf{u}(k+1) - \mathbf{u}(k))\|_2^2 \\
&= \|\mathbf{y} - \mathbf{u}(k)\|_2^2 - 2 (\mathbf{y} - \mathbf{u}(k))^\top (\mathbf{u}(k+1) - \mathbf{u}(k)) + \|\mathbf{u}(k+1) - \mathbf{u}(k)\|_2^2 \\
&= \|\mathbf{y} - \mathbf{u}(k)\|_2^2 - 2\eta (\mathbf{y} - \mathbf{u}(k))^\top \mathbf{H}(k) (\mathbf{y} - \mathbf{u}(k)) \\
&\quad + 2\eta (\mathbf{y} - \mathbf{u}(k))^\top \mathbf{H}(k)^\perp (\mathbf{y} - \mathbf{u}(k)) - 2 (\mathbf{y} - \mathbf{u}(k))^\top \mathbf{I}_2 \\
&\quad + \|\mathbf{u}(k+1) - \mathbf{u}(k)\|_2^2 \\
&\leq (1 - \eta\lambda_0 + \frac{2C\eta n^2 R}{\delta} + \frac{2C\eta n^{3/2} R}{\delta} + \eta^2 n^2) \|\mathbf{y} - \mathbf{u}(k)\|_2^2 \\
&\leq (1 - \frac{\eta\lambda_0}{2}) \|\mathbf{y} - \mathbf{u}(k)\|_2^2.
\end{aligned}$$

The third equality we used the decomposition of $\mathbf{u}(k+1) - \mathbf{u}(k)$. The first inequality we used the Lemma 3.2, the bound on the step size, the bound on $\mathbf{I}_2$, the bound on $\left\| \mathbf{H}(k)^\perp \right\|_2$ and the bound on $\|\mathbf{u}(k+1) - \mathbf{u}(k)\|_2^2$. The last inequality we used the bound of the step size and the bound of $R$. Therefore Condition 4.1 holds for $k' = k+1$. Now by induction, we prove Theorem 4.1. $\qquad \square$

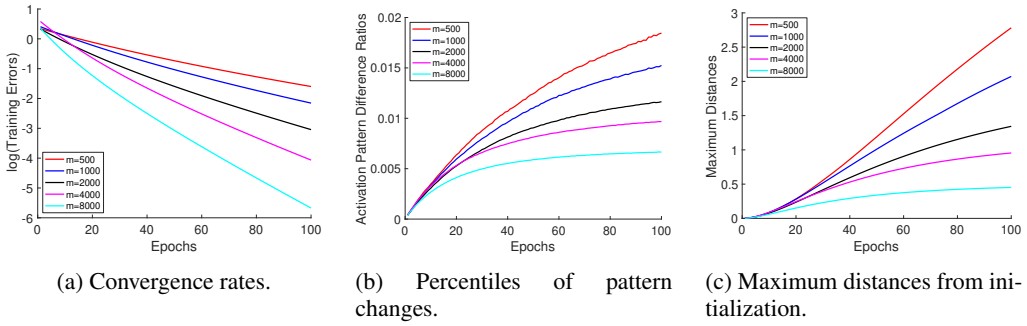

(a) Convergence rates.  (b) Percentiles of pattern changes.  (c) Maximum distances from initialization.

Figure 1: Results on synthetic data.

## 5 EXPERIMENTS

In this section, we use synthetic data to corroborate our theoretical findings. We use the initialization and training procedure described in Section 1. For all experiments, we run 100 epochs of gradient descent and use a fixed step size. We uniformly generate $n = 1000$ data points from a $d = 1000$ dimensional unit sphere and generate labels from a one-dimensional standard Gaussian distribution.

We test three metrics with different widths ($m$). First, we test how the amount of over-parameterization affects the convergence rates. Second, we test the relation between the amount of over-parameterization and the number of pattern changes. Formally, at a given iteration $k$, we check $\frac{\sum_{i=1}^{m}\sum_{r=1}^{m}\mathbb{I}\{\mathrm{sign}(\mathbf{w}_r(0)^\top\mathbf{x}_i)\neq\mathrm{sign}(\mathbf{w}_r(k)^\top\mathbf{x}_i)\}}{mn}$ (there are $mn$ patterns). This aims to verify Lemma 3.2. Last, we test the relation between the amount of over-parameterization and the maximum of the distances between weight vectors and their initializations. Formally, at a given iteration $k$, we check $\max_{r\in[m]}\|\mathbf{w}_r(k) - \mathbf{w}_r(0)\|_2$. This aims to verify Lemma 3.3 and Corollary 4.1.

Figure 1a shows as $m$ becomes larger, we have better convergence rate. We believe the reason is as $m$ becomes larger, $\mathbf{H}(t)$ matrix becomes more stable, and thus has larger least eigenvalue. Figure 1b and Figure 1c show as $m$ becomes larger, the percentiles of pattern changes and the maximum distance from the initialization become smaller. These empirical findings are consistent with our theoretical results.

## 6 CONCLUSION AND DISCUSSION

In this paper we show with over-parameterization, gradient descent provable converges to the global minimum of the empirical loss at a linear convergence rate. The key proof idea is to show the over-parameterization makes Gram matrix remain positive definite for all iterations, which in turn guarantees the linear convergence. Here we list some future directions.

First, we believe our approach can be generalized to deep neural networks. We elaborate the main idea here for gradient flow. Consider a deep neural network of the form

$$f(\mathbf{x}, \mathbf{W}, \mathbf{a}) = \mathbf{a}^\top \sigma\left(\mathbf{W}^{(H)} \cdots \sigma\left(\mathbf{W}^{(1)}\mathbf{x}\right)\right)$$

where $\mathbf{x} \in \mathbb{R}^d$ is the input, $\mathbf{W}^{(1)} \in \mathbb{R}^{m\times d}$ is the first layer, $\mathbf{W}^{(h)} \in \mathbb{R}^{m\times m}$ for $h = 2,\ldots,H$ are the middle layers and $\mathbf{a} \in \mathbb{R}^m$ is the output layer. Recall $u_i$ is the $i$-th prediction. If we use the quadratic loss, we can compute

$$\frac{d\mathbf{W}^{(h)}(t)}{dt} = -\frac{\partial L(\mathbf{W}(t))}{\partial \mathbf{W}^{(h)}(t)} = \sum_{i=1}^{n}(y_i - u_i(t))\frac{\partial u_i(t)}{\partial \mathbf{W}^{(h)}(t)}.$$

Similar to Equation (5), we can calculate

$$\frac{du_i(t)}{dt} = \sum_{h=1}^{H}\langle\frac{\partial u_i(t)}{\partial \mathbf{W}^{(h)}}, \frac{\partial \mathbf{W}^{(h)}}{dt}\rangle = \sum_{j=1}^{n}(y_j - u_j(t))\sum_{h=1}^{H}\mathbf{G}_{ij}^{(h)}(t)$$

where $\mathbf{G}^{(h)}(t) \in \mathbb{R}^{n \times n}$ with $\mathbf{G}_{ij}^{(h)}(t) = \langle \frac{\partial u_i(t)}{\mathbf{W}^{(h)}(t)}, \frac{\partial u_j(t)}{\mathbf{W}^{(h)}(t)} \rangle$. Therefore, similar to Equation (7), we can write

$$\frac{d\mathbf{u}(t)}{dt} = \sum_{h=1}^{H} \mathbf{G}^{(h)}(t)\left(\mathbf{y} - \mathbf{u}(t)\right).$$

Note for every $h \in [H]$, $\mathbf{G}^{(h)}$ is a Gram matrix and thus it is positive semidefinite. If $\sum_{h=1}^{H} \mathbf{G}^{(h)}(t)$ has a lower bounded least eigenvalue for all $t$, then similar to Section 3, gradient flow converges to zero training loss at a linear convergence rate. Based on our observations in Remark 3.1, we conjecture that if $m$ is large enough, $\sum_{h=1}^{H} \mathbf{G}^{(h)}(0)$ is close to a fixed matrix $\sum_{h=1}^{H} \mathbf{G}_{\infty}^{(h)}$ and $\sum_{h=1}^{H} \mathbf{G}^{(h)}(t)$ is close its initialization $\sum_{h=1}^{H} \mathbf{G}^{(h)}(0)$ for all $t > 0$. Therefore, using the same arguments as we used in Section 3, as long as $\sum_{h=1}^{H} \mathbf{G}_{\infty}^{(h)}$ has a lower bounded least eigenvalue, gradient flow converges to zero training loss at a linear convergence rate.

Second, we believe the number of hidden nodes $m$ required can be reduced. For example, previous work (Soudry and Carmon, 2016) showed $m \geq \frac{n}{d}$ is enough to make all differentiable local minima global. In our setting, using advanced tools from probability and matrix perturbation theory to analyze $\mathbf{H}(t)$, we may be able to tighten the bound.

Lastly, in our paper, we used the empirical loss as a potential function to measure the progress. If we use another potential function, we may be able to prove the convergence rates of accelerated methods. This technique has been exploited in Wilson et al. (2016) for analyzing convex optimization. It would be interesting to bring their idea to analyze other first order methods for optimizing neural networks.

ACKNOWLEDGMENTS

This research was partly funded by AFRL grant FA8750-17-2-0212 and DARPA D17AP00001. We thank Wei Hu, Jason D. Lee and Ruosong Wang for useful discussions.

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

# A    TECHNICAL PROOFS FOR SECTION 3

*Proof of Theorem 3.1.* The proof of this lemma just relies on standard real and functional analysis. Let $\mathcal{H}$ be the Hilbert space of integrable $d$-dimensional vector fields on $\mathbb{R}^d$: $f \in \mathcal{H}$ if $\mathbb{E}_{\mathbf{w} \sim N(\mathbf{0}, \mathbf{I})} \left[ |f(\mathbf{w})|^2 \right] < \infty$. The inner product of this space is then $\langle f, g \rangle_{\mathcal{H}} = \mathbb{E}_{\mathbf{w} \sim N(\mathbf{0}, \mathbf{I})} \left[ f(\mathbf{w})^\top g(\mathbf{w}) \right]$.

ReLU activation induces an infinite-dimensional feature map $\phi$ which is defined as for any $\mathbf{x} \in \mathbb{R}^d$, $(\phi(\mathbf{x}))(\mathbf{w}) = \mathbf{x} \mathbb{I} \left\{ \mathbf{w}^\top \mathbf{x} \geq 0 \right\}$ where $\mathbf{w}$ can be viewed as the index. Now to prove $\mathbf{H}^\infty$ is strictly positive definite, it is equivalent to show $\phi(\mathbf{x}_1), \ldots, \phi(\mathbf{x}_n) \in \mathcal{H}$ are linearly independent. Suppose that there are $\alpha_1, \cdots, \alpha_n \in \mathbb{R}$ such that

$$\alpha_1 \phi(\mathbf{x}_i) + \cdots + \alpha_n \phi(\mathbf{x}_n) = 0 \text{ in } \mathcal{H}.$$

This means that

$$\alpha_1 \phi(\mathbf{x}_1)(\mathbf{w}) + \cdots + \alpha_n \phi(\mathbf{x}_n)(\mathbf{w}) = 0 \text{ a.e.}$$

Now we prove $\alpha_i = 0$ for all $i$.

We define $D_i = \left\{ \mathbf{w} \in \mathbb{R}^d : \mathbf{w}^\top \mathbf{x}_i = 0 \right\}$. This is set of discontinuities of $\phi(\mathbf{x}_i)$. The following lemma characterizes the basic property of these discontinuity sets.

**Lemma A.1.** *If for any $i \neq j$, $\mathbf{x}_i \not\parallel \mathbf{x}_j$, then for any $i \in [m]$, $D_i \not\subset \bigcup_{j \neq i} D_j$.*

Now for a fixed $i \in [n]$, since $D_i \not\subset \bigcup_{j \neq i} D_j$, we can choose $\mathbf{z} \in D_i \setminus \bigcup_{j \neq i} D_j$. Note $D_j, j \neq i$ are closed sets. We can pick $r_0 > 0$ small enough such that $B(\mathbf{z}, r) \cap D_j = \emptyset, \forall j \neq i, r \leq r_0$. Let $B(\mathbf{z}, r) = B_r^+ \sqcup B_r^-$ where

$$B_r^+ = B(\mathbf{z}, r) \cap \left\{ \mathbf{w} \in \mathbb{R}^d : \mathbf{w}^\top \mathbf{x}_i > 0 \right\}.$$

For $j \neq i$, $\phi(\mathbf{x}_j)(\mathbf{w})$ is continuous in a neighborhood of $\mathbf{z}$, then for any $\epsilon > 0$ there is a small enough $r > 0$ such that

$$\forall \mathbf{w} \in B(\mathbf{z}, r), |\phi(\mathbf{x}_j)(\mathbf{w}) - \phi(\mathbf{x}_j)(\mathbf{z})| < \epsilon.$$

Let $\mu$ be the Lebesgue measure on $\mathbb{R}^d$. We have

$$\left| \frac{1}{\mu(B_r^+)} \int_{B_r^+} \phi(\mathbf{x}_j)(\mathbf{w}) d\mathbf{w} - \phi(\mathbf{x}_j)(\mathbf{z}) \right| \leq \frac{1}{\mu(B_r^+)} \int_{B_r^+} |\phi(\mathbf{x}_j)(\mathbf{w}) - \phi(\mathbf{x}_j)(\mathbf{z})| \, d\mathbf{w} < \epsilon$$

and similarly

$$\left| \frac{1}{\mu(B_r^-)} \int_{B_r^-} \phi(\mathbf{x}_j)(\mathbf{w}) d\mathbf{w} - \phi(\mathbf{x}_j)(\mathbf{z}) \right| \leq \frac{1}{\mu(B_r^-)} \int_{B_r^-} |\phi(\mathbf{x}_j)(\mathbf{w}) - \phi(\mathbf{x}_j)(\mathbf{z})| \, d\mathbf{w} < \epsilon.$$

Thus, we have

$$\lim_{r \to 0+} \frac{1}{\mu(B_r^+)} \int_{B_r^+} \phi(\mathbf{x}_j)(\mathbf{w}) d\mathbf{w} = \lim_{r \to 0+} \frac{1}{\mu(B_r^-)} \int_{B_r^-} \phi(\mathbf{x}_j)(\mathbf{w}) d\mathbf{w} = \phi(\mathbf{x}_j)(\mathbf{z}).$$

Therefore, as $r \to 0+$, by continuity, we have

$$\forall j \neq i, \frac{1}{\mu(B_r^+)} \int_{B_r^+} \phi(\mathbf{x}_j)(\mathbf{w}) d\mathbf{w} - \frac{1}{\mu(B_r^-)} \int_{B_r^-} \phi(\mathbf{x}_j)(\mathbf{w}) d\mathbf{w} \to 0 \tag{9}$$

Next recall that $(\phi(\mathbf{x}))(\mathbf{w}) = \mathbf{x} \mathbb{I} \left\{ \mathbf{x}^\top \mathbf{w} > 0 \right\}$, so for $\mathbf{w} \in B_r^+$ and $\mathbf{x}_i$, $(\phi(\mathbf{x}_i))(\mathbf{w}) = \mathbf{x}_i$. Then, we have

$$\lim_{r \to 0+} \frac{1}{\mu(B_r^+)} \int_{B_r^+} \phi(\mathbf{x}_j)(\mathbf{w}) d\mathbf{w} = \lim_{r \to 0+} \frac{1}{\mu(B_r^+)} \int_{B_r^+} \mathbf{x}_i d\mathbf{w} = \mathbf{x}_i. \tag{10}$$

For $\mathbf{w} \in B_r^-$ and $\mathbf{x}_i$, we know $(\phi(\mathbf{x}_i))(\mathbf{w}) = 0$. Then we have

$$\lim_{r \to 0+} \frac{1}{\mu(B_r^-)} \int_{B_r^-} \phi(\mathbf{x}_i)(\mathbf{w}) d\mathbf{w} = \lim_{r \to 0+} \frac{1}{\mu(B_r^-)} \int_{B_r^-} 0 d\mathbf{w} = 0 \tag{11}$$

Now recall $\sum_i \alpha_i \phi(\mathbf{x}_i) \equiv 0$. Using Equation (9), (10) and (11), we have

$$
\begin{aligned}
0 &= \lim_{r\to 0+} \frac{1}{\mu(B_r^+)} \int_{B_r^+} \sum_j \alpha_j \phi(\mathbf{x}_j)(\mathbf{w}) d\mathbf{w} - \lim_{r\to 0+} \frac{1}{\mu(B_r^-)} \int_{B_r^-} \sum_j \alpha_j \phi(\mathbf{x}_j)(\mathbf{w}) d\mathbf{w} \\
&= \sum_j \alpha_j \left( \lim_{r\to 0+} \frac{1}{\mu(B_r^+)} \int_{B_r^+} \phi(\mathbf{x}_j)(\mathbf{w}) d\mathbf{w} - \lim_{r\to 0+} \frac{1}{\mu(B_r^-)} \int_{B_r^-} \phi(\mathbf{x}_j)(\mathbf{w}) d\mathbf{w} \right) \\
&= \sum_j \alpha_j \left( \delta_{ij} \mathbf{x}_i \right) \\
&= \alpha_i \mathbf{x}_i
\end{aligned}
$$

Since $\mathbf{x}_i \neq 0$, we must have $\alpha_i = 0$. We complete the proof. $\qquad\square$

*Proof of Lemma A.1.* Let $\mu$ be the canonical Lebesgue measure on $D_i$. We have $\sum_{j\neq i} \mu(D_i \cap D_j) = 0$ because $D_i \cap D_j$ is a hyperplane in $D_i$. Now we bound

$$
\mu(D_i \cap \bigcup_{j\neq i} D_j) \leq \sum_{j\neq i} \mu(D_i \cap D_j) = 0.
$$

This implies our desired result. $\qquad\square$

*Proof of Lemma 3.1.* For every fixed $(i,j)$ pair, $\mathbf{H}_{ij}(0)$ is an average of independent random variables. Therefore, by Hoeffding inequality, we have with probability $1 - \delta'$,

$$
\left| \mathbf{H}_{ij}(0) - \mathbf{H}_{ij}^\infty \right| \leq \frac{2\sqrt{\log(1/\delta')}}{\sqrt{m}}.
$$

Setting $\delta' = n^2 \delta$ and applying union bound over $(i,j)$ pairs, we have for every $(i,j)$ pair with probability at least $1 - \delta$

$$
\left| \mathbf{H}_{ij}(0) - \mathbf{H}_{ij}^\infty \right| \leq \frac{4\sqrt{\log(n/\delta)}}{\sqrt{m}}.
$$

Thus we have

$$
\|\mathbf{H}(0) - \mathbf{H}^\infty\|_2^2 \leq \|\mathbf{H}(0) - \mathbf{H}^\infty\|_F^2 \leq \sum_{i,j} \left| \mathbf{H}_{ij}(0) - \mathbf{H}_{ij}^\infty \right|^2 \leq \frac{16n^2 \log(n/\delta)}{m}.
$$

Thus if $m = \Omega\left( \frac{n^2 \log(n/\delta)}{\lambda_0^2} \right)$ we have the desired result. $\qquad\square$

*Proof of Lemma 3.4.* Suppose the conclusion does not hold at time $t$. If there exists $r \in [m]$, $\|\mathbf{w}_r(t) - \mathbf{w}_r(0)\| \geq R'$ or $\|\mathbf{y} - \mathbf{u}(t)\|_2^2 > \exp(-\lambda_0 t) \|\mathbf{y} - \mathbf{u}(0)\|_2^2$, then by Lemma 3.3 we know there exists $s \leq t$ such that $\lambda_{\min}(\mathbf{H}(s)) < \frac{1}{2}\lambda_0$. By Lemma 3.2 we know there exists

$$
t_0 = \inf \left\{ t > 0 : \max_{r\in[m]} \|\mathbf{w}_r(t) - \mathbf{w}_r(0)\|_2^2 \geq R \right\}.
$$

Thus at $t_0$, there exists $r \in [m]$, $\|\mathbf{w}_r(t_0) - \mathbf{w}_r(0)\|_2^2 = R$. Now by Lemma 3.2, we know $\mathbf{H}(t_0) \geq \frac{1}{2}\lambda_0$ for $t' \leq t_0$. However, by Lemma 3.3, we know $\|\mathbf{w}_r(t_0) - \mathbf{w}_r(0)\|_2 < R' < R$. Contradiction.

For the other case, at time $t$, $\lambda_{\min}(\mathbf{H}(t)) < \frac{1}{2}\lambda_0$ we know there exists

$$
t_0 = \inf \left\{ t \geq 0 : \max_{r\in[m]} \|\mathbf{w}_r(t) - \mathbf{w}_r(0)\|_2^2 \geq R \right\}.
$$

The rest of the proof is the same as the previous case. $\qquad\square$

## A.1 PROOF OF THEOREM 3.3

In this section we show using gradient flow to jointly train both the first layer and the output layer we can still achieve $0$ training loss. We follow the same approach we used in Section 3. Recall the gradient for $\mathbf{a}$.

$$\frac{\partial L(\mathbf{w}, \mathbf{a})}{\partial \mathbf{a}} = \frac{1}{\sqrt{m}} \sum_{i=1}^{n} \left( f\left(\mathbf{w}, \mathbf{a}, \mathbf{x}_i\right) - y_i \right) \begin{pmatrix} \sigma\left(\mathbf{w}_1^\top \mathbf{x}_i\right) \\ \cdots \\ \sigma\left(\mathbf{w}_m^\top \mathbf{x}_i\right) \end{pmatrix}. \tag{12}$$

We compute the dynamics of an individual prediction.

$$\frac{du_i(t)}{dt} = \sum_{r=1}^{m} \langle \frac{\partial u_i(t)}{\partial \mathbf{w}_r(t)}, \frac{\partial \mathbf{w}_r(t)}{\partial t} \rangle + \sum_{r=1}^{m} \frac{du_i(t)}{da_r(t)} \cdot \frac{da_r(t)}{dt}. \tag{13}$$

Recall we have found a convenient expression for the first term.

$$\sum_{r=1}^{m} \langle \frac{\partial u_i(t)}{\partial \mathbf{w}_r(t)}, \frac{\partial \mathbf{w}_r(t)}{\partial t} \rangle = \sum_{j=1}^{n} \left(y_j - u_j(t)\right) \mathbf{H}_{ij}(t)$$

where

$$\mathbf{H}_{ij}(t) = \frac{1}{m} \mathbf{x}_i^\top \mathbf{x}_j \sum_{r=1}^{m} a_r^2(t) \mathbb{I}\left\{\mathbf{x}_i^\top \mathbf{w}_r(t) \geq 0, \mathbf{x}_j^\top \mathbf{w}_r(t) \geq 0\right\}.$$

For the second term, it easy to derive

$$\sum_{r=1}^{m} \frac{du_i(t)}{d\mathbf{a}_r} \cdot \frac{d\mathbf{a}_r(t)}{dt} = \sum_{r=1}^{m} \left(y_j - u_j(t)\right) \mathbf{G}_{ij}(t)$$

where

$$\mathbf{G}_{ij}(t) = \frac{1}{m} \sigma\left(\mathbf{w}_r^\top \mathbf{x}_i\right) \sigma\left(\mathbf{w}_r^\top \mathbf{x}_j\right). \tag{14}$$

Therefore we have

$$\frac{d\mathbf{u}(t)}{dt} = \left(\mathbf{H}(t) + \mathbf{G}(t)\right)\left(\mathbf{y} - \mathbf{u}(t)\right).$$

First use the same concentration arguments as in Lemma 3.1, we can show $\lambda_{\min}(\mathbf{H}(0)) \geq \frac{3\lambda_0}{4}$ with $1 - \delta$ probability over the initialization. In the following, our arguments will base on that $\lambda_{\min}(\mathbf{H}(0)) \geq \frac{3\lambda_0}{4}$.

The following lemma shows as long as $\mathbf{H}(t)$ has lower bounded least eigenvalue, gradient flow enjoys a linear convergence rate. The proof is analogue to the first part of the proof of Lemma 3.3.

**Lemma A.2.** *If for* $0 \leq s \leq t$, $\lambda_{\min}(\mathbf{H}(s)) \geq \frac{\lambda_0}{2}$, *we have* $\|\mathbf{y} - \mathbf{u}(t)\|_2^2 \leq \exp(-\lambda_0 t) \|\mathbf{y} - \mathbf{u}(0)\|_2^2$.

*Proof of Lemma A.2.* We can calculate the loss function dynamics

$$\frac{d}{dt} \|\mathbf{y} - \mathbf{u}(t)\|_2^2 = -2\left(\mathbf{y} - \mathbf{u}(t)\right)^\top \left(\mathbf{H}(t) + \mathbf{G}(t)\right)\left(\mathbf{y} - \mathbf{u}(t)\right)$$

$$\leq -2\left(\mathbf{y} - \mathbf{u}(t)\right)^\top \left(\mathbf{H}(t)\right)\left(\mathbf{y} - \mathbf{u}(t)\right)$$

$$\leq -\lambda_0 \|\mathbf{y} - \mathbf{u}(t)\|_2^2$$

where in the first inequality we use the fact that $\mathbf{G}(t)$ is Gram matrix thus it is positive.[8] Thus we have $\frac{d}{dt}\left(\exp(\lambda_0 t) \|\mathbf{y} - \mathbf{u}(t)\|_2^2\right) \leq 0$ and $\exp(\lambda_0 t) \|\mathbf{y} - \mathbf{u}(t)\|_2^2$ is a decreasing function with respect to $t$. Using this fact we can bound the loss

$$\|\mathbf{y} - \mathbf{u}(t)\|_2^2 \leq \exp(-\lambda_0 t) \|\mathbf{y} - \mathbf{u}(0)\|_2^2.$$

$\square$

---

[8] In the proof, we have not take the advantage of $\mathbf{G}(t)$ being a positive semidefinite matrix. Note if $\mathbf{G}(t)$ is strictly positive definite, we can achieve faster convergence rate.

We continue to follow the analysis in Section 3. For convenience, we define

$$R_w = \frac{\sqrt{2\pi}\lambda_0\delta}{32n^2}, R_a = \frac{\lambda_0}{16n^2}, R'_w = \frac{4\sqrt{n}\,\|\mathbf{y} - \mathbf{u}_0\|_2}{\sqrt{m}\lambda_0}, R'_a = \frac{8\sqrt{n}\,\|\mathbf{y} - \mathbf{u}(0)\|_2\,\sqrt{\log(mn/\delta)}}{\sqrt{m}\lambda_0}.$$

The first lemma characterizes how the perturbation in $\mathbf{a}$ and $\mathbf{W}$ affect the Gram matrix.

**Lemma A.3.** *With probability at least $1 - \delta$ over initialization, if a set of weight vectors $\{\mathbf{w}_r\}_{r=1}^m$ and the output weight $\mathbf{a}$ satisfy for all $r \in [m]$, $\|\mathbf{w}_r - \mathbf{w}_r(0)\|_2 \leq R_w$ and $|a_r - a_r(0)| \leq R_a$, then the matrix $\mathbf{H} \in \mathbb{R}^{n \times n}$ defined by*

$$\mathbf{H}_{ij} = \frac{1}{m}\mathbf{x}_i^\top \mathbf{x}_j \sum_{r=1}^m \mathbf{a}_r^2 \mathbb{I}\left\{\mathbf{w}_r^\top \mathbf{x}_i \geq 0, \mathbf{w}_r^\top \mathbf{x}_j \geq 0\right\}$$

*satisfies $\|\mathbf{H} - \mathbf{H}(0)\|_2 \leq \frac{\lambda_0}{4}$ and $\lambda_{\min}(\mathbf{H}) > \frac{\lambda_0}{2}$.*

*Proof of Lemma A.3.* Define a surrogate Gram matrix,

$$\mathbf{H}' = \frac{1}{m}\sum_{r=1}^m \mathbb{I}\left\{\mathbf{w}_r^\top \mathbf{x}_i \geq 0, \mathbf{w}_r^\top \mathbf{x}_j \geq 0\right\}.$$

Using the same analysis for Lemma 3.3, we know $\|\mathbf{H}' - \mathbf{H}(0)\|_2 \leq \frac{4n^2 R_w}{\sqrt{2\pi}\delta}$. Now we bound $\mathbf{H} - \mathbf{H}'$. For fixed $(i,j) \in [n] \times [n]$, we have

$$\mathbf{H}_{ij} - \mathbf{H}'_{ij} = \left|\mathbf{x}_i^\top \mathbf{x}_j \frac{1}{m}\sum_{r=1}^m (\mathbf{a}_r^2 - 1)\mathbb{I}\left\{\mathbf{w}_r^\top \mathbf{x}_i \geq 0, \mathbf{w}_r^\top \mathbf{x}_j \geq 0\right\}\right| \leq \max_{r\in[m]}\left|\mathbf{a}_r^2 - 1\right| \leq 2R_a.$$

Therefore, we have $\|\mathbf{H} - \mathbf{H}'\|_2 \leq \sum_{(i,j)\in[n]\times[n]}\left|\mathbf{H}_{ij} - \mathbf{H}'_{ij}\right| \leq 2n^2 R_a$. Combing these two inequalities we have $\|\mathbf{H} - \mathbf{H}(0)\|_2 \leq \frac{4n^2 R_w}{\sqrt{2\pi}\delta} + 2n^2 R_a \leq \frac{\lambda_0}{4}$. $\qquad\square$

**Lemma A.4.** *Suppose for $0 \leq s \leq t$, $\lambda_{\min}(\mathbf{H}(s)) \geq \frac{\lambda_0}{2}$ and $|\mathbf{a}_r(s) - \mathbf{a}_r(0)| \leq R_a$. Then we have $\|\mathbf{w}_r(t) - \mathbf{w}_r(0)\|_2 \leq R'_w$.*

*Proof of Lemma A.4.* We bound the gradient. Recall for $0 \leq s \leq t$,

$$\left\|\frac{d}{dt}\mathbf{w}_r(s)\right\|_2 = \left\|\sum_{i=1}^n (y_i - u_i)\frac{1}{\sqrt{m}}a_r(t)\mathbf{x}_i \mathbb{I}\left\{\mathbf{w}_r(t)^\top \mathbf{x}_i \geq 0\right\}\right\|_2$$

$$\leq \frac{1}{\sqrt{m}}\sum_{i=1}^n |y_i - u_i(s)|\,|a_r(0) + R_a|$$

$$\leq \frac{2\sqrt{n}}{\sqrt{m}}\|\mathbf{y} - \mathbf{u}(s)\|_2 \leq \frac{2\sqrt{n}}{\sqrt{m}}\exp(-\lambda_0 s/2)\|\mathbf{y} - \mathbf{u}(0)\|_2.$$

Integrating the gradient and using Lemma A.2, we can bound the distance from the initialization

$$\|\mathbf{w}_r(t) - \mathbf{w}_r(0)\|_2 \leq \int_0^t \left\|\frac{d}{ds}\mathbf{w}_r(s)\right\|_2 ds \leq \frac{4\sqrt{n}\,\|\mathbf{y} - \mathbf{u}(0)\|_2}{\sqrt{m}\lambda_0}.$$

$\qquad\square$

**Lemma A.5.** *With probability at least $1 - \delta$ over initialization, the following holds. Suppose for $0 \leq s \leq t$, $\lambda_{\min}(\mathbf{H}(s)) \geq \frac{\lambda_0}{2}$ and $\|\mathbf{w}_r(s) - \mathbf{w}_r(0)\|_2 \leq R_w$. Then we have $|a(t) - a_r(0)| \leq R'_a$ for all $r \in [m]$.*

*Proof of Lemma A.5.* Note for any $i \in [n]$ and $r \in [m]$, $\mathbf{w}_r(0)^\top \mathbf{x}_i \sim N(0,1)$. Therefore applying Gaussian tail bound and union bound we have with probability at least $1 - \delta$, for all $i \in [n]$ and

$r \in [m]$, $\left| \mathbf{w}_r(0)^\top \mathbf{x}_i \right| \leq 3\sqrt{\log\left(\frac{mn}{\delta}\right)}$. Now we bound the gradient. Recall for $0 \leq s \leq t$,

$$
\left| \frac{d}{ds} a_r(s) \right| = \left| \frac{1}{\sqrt{m}} \sum_{i=1}^{n} \left( f\left( \mathbf{w}(s), \mathbf{a}(s), \mathbf{x}_i \right) - y_i \right) \left( \sigma\left( \mathbf{w}_r(s)^\top \mathbf{x}_i \right) \right) \right|
$$

$$
\leq \frac{1}{\sqrt{m}} \sqrt{n} \left\| \mathbf{y} - \mathbf{u}(s) \right\|_2 \left( \left| \mathbf{w}_r(s)^\top \mathbf{x}_i \right| + R_w \right)
$$

$$
\leq \frac{1}{\sqrt{m}} \sqrt{n} \left\| \mathbf{y} - \mathbf{u}(s) \right\|_2 \left( 3\sqrt{\log\left(\frac{mn}{\delta}\right)} + R_w \right) \leq \frac{4\sqrt{n} \left\| \mathbf{y} - \mathbf{u}(0) \right\| \exp(-\lambda_0 s/2) \sqrt{\log\left( mn/\delta \right)}}{\sqrt{m}}.
$$

Integrating the gradient, we can bound the distance from the initialization

$$
\left\| \mathbf{a}_r(t) - \mathbf{a}_r(0) \right\|_2 \leq \int_0^t \left\| \frac{d}{ds} \mathbf{w}_r(s) \right\|_2 ds \leq \frac{8\sqrt{n} \left\| \mathbf{y} - \mathbf{u}(0) \right\|_2 \sqrt{\log(mn/\delta)}}{\sqrt{m}\lambda_0}.
$$

$\square$

**Lemma A.6.** *If $R'_w < R_w$ and $R'_a < R_a$, we have for all $t \geq 0$, $\lambda_{\min}(\mathbf{H}(t)) \geq \frac{1}{2}\lambda_0$, for all $r \in [m]$, $\left\| \mathbf{w}_r(t) - \mathbf{w}_r(0) \right\|_2 \leq R'_w$, $\left| a_r(t) - a_r(0) \right| \leq R'_a$ and $\left\| \mathbf{y} - \mathbf{u}(t) \right\|_2^2 \leq \exp(-\lambda_0 t) \left\| \mathbf{y} - \mathbf{u}(0) \right\|_2^2$.*

*Proof of Lemma A.6.* We prove by contradiction. Let $t > 0$ be the smallest time that the conclusion does not hold. Then either $\lambda_{\min}(\mathbf{H}(t)) < \frac{\lambda_0}{2}$ or there exists $r \in [m]$, $\left\| \mathbf{w}_r(t) - \mathbf{w}_r(0) \right\|_2 \leq R'_w$ or there exists $r \in [m]$, $\left| a_r - a_r(0) \right| \leq R'_a$. If $\lambda_{\min}(\mathbf{H}(t)) < \frac{\lambda_0}{2}$, by Lemma A.3, we know there exists $s < t$ such that either there exists $r \in [m]$, $\left\| \mathbf{w}_r(s) - \mathbf{w}_r(0) \right\|_2 \leq R_w$ or there exists $r \in [m]$, $\left| a_r(s) - a_r(0) \right| \leq R_a$. However, since $R'_w < R_w$ and $R'_a < R_a$. This contradicts with the minimality of $t$. If there exists $r \in [m]$, $\left\| \mathbf{w}_r - \mathbf{w}_r(0) \right\|_2 \leq R'_w$, then by Lemma A.4, we know there exists $s < t$ such that $r \in [m]$, $\left| a_r(s) - a_r(0) \right| \leq R_a$ or $\lambda_{\min}(\mathbf{H}(s)) < \frac{\lambda_0}{2}$. However, since $R'_w < R_w$ and $R'_a < R_a$. This contradicts with the minimality of $t$. The last case is similar for which we can simply apply Lemma A.5. $\square$

Based on Lemma A.2, we only need to ensure $R'_w < R_w$ and $R_a < R'_a$. By the proof in Section 3, we know with probability at least $\delta$, $\left\| \mathbf{y} - \mathbf{u}(0) \right\|_2 \leq \frac{C\sqrt{n}}{\delta}$ for some large constant $C$. Note our section on $m$ in Theorem 3.3 suffices to ensure $R'_w < R_w$ and $R_a < R'_a$. We now complete the proof.

# B   TECHNICAL PROOFS FOR SECTION 4

*Proof of Corollary 4.1.* We use the norm of gradient to bound this distance.

$$
\left\| \mathbf{w}_r(k+1) - \mathbf{w}_r(0) \right\|_2 \leq \eta \sum_{k'=0}^{k} \left\| \frac{\partial L(\mathbf{W}(k'))}{\partial \mathbf{w}_r(k')} \right\|_2
$$

$$
\leq \eta \sum_{k'=0}^{k} \frac{\sqrt{n} \left\| \mathbf{y} - \mathbf{u}(k') \right\|_2}{\sqrt{m}}
$$

$$
\leq \eta \sum_{k'=0}^{k} \frac{\sqrt{n}(1 - \frac{\eta\lambda}{2})^{k'/2}}{\sqrt{m}} \left\| \mathbf{y} - \mathbf{u}(k') \right\|_2
$$

$$
\leq \eta \sum_{k'=0}^{\infty} \frac{\sqrt{n}(1 - \frac{\eta\lambda_0}{2})^{k'/2}}{\sqrt{m}} \left\| \mathbf{y} - \mathbf{u}(k') \right\|_2
$$

$$
= \frac{4\sqrt{n} \left\| \mathbf{y} - \mathbf{u}(0) \right\|_2}{\sqrt{m}\lambda_0}.
$$

$\square$

*Proof of Lemma 4.1.* For a fixed $i \in [n]$ and $r \in [m]$, by anti-concentration inequality, we know $\mathbb{P}(A_{ir}) \leq \frac{2R}{\sqrt{2\pi}}$. Thus we can bound the size of $S_i^{\perp}$ in expectation.

$$\mathbb{E}\left[\left|S_i^{\perp}\right|\right] = \sum_{r=1}^{m} \mathbb{P}(A_{ir}) \leq \frac{2mR}{\sqrt{2\pi}}. \tag{15}$$

Summing over $i = 1, \ldots, n$, we have

$$\mathbb{E}\left[\sum_{i=1}^{n}\left|S_i^{\perp}\right|\right] \leq \frac{2mnR}{\sqrt{2\pi}}.$$

Thus by Markov's inequality, we have with probability at least $1 - \delta$

$$\sum_{i=1}^{n}\left|S_i^{\perp}\right| \leq \frac{CmnR}{\delta}. \tag{16}$$

for some large positive constant $C > 0$. $\qquad\square$

