# OpenReview forum: "Gradient Descent Provably Optimizes Over-parameterized Neural Networks"
_ICLR.cc/2019/Conference_

### Official Review · AnonReviewer2 · 2018-10-31
**An elegant proof on convergence of gradient descent for over-parameterized two-layer ReLU neural networks**

**Rating:** 7
**Confidence:** 4

**Review:**

This paper studies convergence of gradient descent on a two-layer fully connected ReLU network with binary output and square loss. The main result is that if the number of hidden units is polynomially large in terms of the number of training samples, then under suitable randomly initialization conditions and given that the output weights are fixed, gradient descent necessarily converge to zero training loss.

Pros:
The paper is presented clearly enough, but I still urge the authors to carefully check for typos and grammatical mistakes as they revise the paper. As far as I have checked, the proofs are correct. The analysis is quite simple and elegant. This is one thing that I really like about this paper compared to previous work.

Cons:
The current setting and conditions for the main result to hold are quite a bit limited. If one has polynomially large number of neurons (i.e. on the order of n^6 where n is number of training samples) as stated in the paper, then the weights of the hidden layer can be easily chosen so that the outputs of all training samples become linearly independent in the hidden layer (see e.g. [1] for the construction, which requires only n neurons even with weight sharing) , and thus fixing these weights and optimizing for the output weights would lead directly to a convex problem with the same theoretical guarantee. At this point, it would be good to explain why this paper is focusing on the opposite setting, namely fixing the output weights and learning just the hidden layer weights, because it seems that this just makes the problem become more non-trivial compared to the previous case while yielding almost the same results . Either way, this is not the way how practical neural networks are trained as only a subset of the weights are optimized. Thus it's hard to conclude from here why the commonly used GD w.r.t. all variables converges to zero loss as stated in the abstract.

The condition on the Gram matrix H_infty in Theorem 3.1 seems to be critical. I would like to see the proof that this condition can be fulfilled under certain conditions on the training data.

In Lemma 3.1, it seems that "log^2(n/delta)" should be "log(n^2/delta)"?

Despite the above limitations, I think that the analysis in this paper is still interesting (mainly due to its simplicity) from a theoretical perspective. Given the difficulty of the problem, I'm happy to vote for its acceptance.

[1] Optimization landscape and expressivity of deep CNNs

---

> ### Author Response · Authors · 2018-11-19
> **Response to Reviewer 2**
>
> We thank for your careful and encouraging review.  We believe our revised version has addressed most of your concerns.
>
> 1.    We have added discussions on the problem of fixing the first layer and only training the output layer in footnote 3. We believe the learned function is different from the function learned by fixing the output layer and only training the first layer. We would also like to point out that many previous papers considered the same setting but did not rigorously prove the global convergence of gradient descent.
>
> 2.    We have added a new theorem (Theorem 3.3) which shows applying gradient flow to optimize all variables still enjoys a linear convergence rate. To prove Theorem 3.3, we use the same arguments as we used to prove Theorem 3.1 with slightly more calculations. Therefore, we have shown analyzing the case that both layers are trained is just as hard as analyzing the case where only the first layer is trained.
>
> 3.    We have added a new theorem (Theorem 3.1) which shows as long as no two inputs are parallel, H^{\infty} is non-degenerate.
>
> We thank the reviewer again. We welcome all further comments!

---

### Official Review · AnonReviewer1 · 2018-11-02
**Interesting result on optimization of two-layer network with ReLU activations**

**Rating:** 8
**Confidence:** 4

**Review:**

This work considers optimizing a two-layer over-parameterized ReLU network with the squared loss and given a data set with arbitrary labels. It is shown that for a sufficiently large number of hidden neurons (polynomially in number of samples) gradient descent converges to a global minimum with a linear convergence rate. The proof idea is to show that a certain Gram matrix of the data, which depends also on the weights, has a lower bounded minimum eigenvalue throughout the optimization process. Then, it is shown that this property implies convergence of gradient descent.

This work is very interesting. Proving convergence of gradient descent for over-parameterized networks with ReLU activations and data with arbitrary labels is a major challenge. It is surprising that the authors found a relatively concise proof in the case of two-layer networks. The insight on the connection between the spectral properties of the Gram matrix and convergence of gradient descent is nice and seems to be a very promising technique for future work. One weakness of the result is the extremely large number of hidden neurons that are required to guarantee convergence.

The paper is clearly written in most parts. The statement of Lemma 3.2 and its application appear to be incorrect as mentioned in the comments. I am convinced by the authors' response and the current proof that it can be fixed by defining an event which is independent of t. Moreover, I think it would be nice to include experiments that corroborate the theoretical findings. Specifically, it would be interesting to see if in practice most of the patterns of ReLUs do not change or if there is some other phenomenon.

As mentioned in the comments, it would be good to add a discussion on the assumption of non-degeneracy of the H^{infty} matrix and include a proof (or exact reference) which shows under which conditions the minimum eigenvalue is positive.

-------------Revision--------------

I disagree with most of the points that AnonReviewer3 raised (e.g., second layer fixed is not hard, contribution is limited). I do agree that the main weakness is the number of neurons.  However, I think that the result is significant nonetheless. I did not change my original score.

---

> ### Author Response · Authors · 2018-11-19
> **Response to Reviewer 1**
>
> We thank for your encouraging review.
> We have modified our paper according to your suggestions:
> •	We fixed lemma 3.2.
> •	We added a new theorem (Theorem 3.1) showing the non-degeneracy of H^{\infty} matrix.
> •	We also added some experiments to corroborate our theoretical findings. Indeed, most of the patterns of ReLUs do not change. Furthermore, over-parameterization leads to faster convergence rate.
>
> We thank the reviewer again. We welcome all further comments!

---

### Official Review · AnonReviewer4 · 2018-11-11
**Interesting paper studying gradient descent in over-parameterized simple NNs**

**Rating:** 8
**Confidence:** 4

**Review:**

This paper studies one hidden layer neural networks with square loss, where they show that in over-parameterized setting, random initialization + gradient descent gets to zero loss. The results depend on the property of data matrix, but not the output values.

The high level idea of the proof is quite different from recent papers, and it would be quite interesting to see how powerful this is for deep neural nets, and whether any insights could help practitioners in the future.

Some discussions regarding the results:

I would suggest the authors to be specific about ‘with high probability’, whether it is 1-c, or 1-n^{-c}. The proof step using Markov’s inequality gives 1-c probability, which is stated as ‘with high probability’. What about other ‘high probability’ statements?

In the statement of Theorem 3.1 and 4.1, please add ‘i.i.d.’ (independence) for generating w_r s.

The current statement of Lemma 3.2 is confusing. The authors state that given t, w.h.p. (let’s say 0.9 for now) over initialization, the minimum eigenvalue is lower bounded. This does not imply, for example, that there exists an initialization, such that for 20 different t s, the minimum eigenvalue is lower bounded. The proof uses Markov’s inequality for a single t. Therefore, I am slightly worried about its correctness. I hope the authors could address my concern.

Also, in the proof of Lemma 3.2, (just to improve the readability,) I would suggest the authors to make it clear that the expectation is taken over the initialization of the weights.

Some typos:

‘converges’ -> ‘converges to’ in the abstract
‘close’ -> ‘close to’ on page 5
‘a crucial’ -> ‘a crucial role’ on page 5
In the proof of Lemma 3.2, x_0 should be x_i
whether using boldface for H_{ij} should be consistent
'The next lemma shows we show' in page 6
'Markov inequality' -> ‘Markov’s inequality’
‘a fixed a neural network architecture’ in page 8

It is good to see other comments and discussions on this paper. I believe the authors will make a revision and I would be happy to see the new version of the paper and re-evaluate if some of my comments are not correct.

---

> ### Author Response · Authors · 2018-11-19
> **Response to Reviewer 4**
>
> We thank for your careful review.
>
> We have modified our draft according to your suggestions:
> •    We changed the statement of Lemma 3.2, and now it is independent of t.
> •    We have added more discussions on how to generalize our technique to analyze multiple layers. In the conclusion section, we have described a concrete plan for analysis.
> •    For all theorems and lemmas, we have added failure probability and how the amount of over-parameterization depends on this failure probability.
> •    We have fixed the typos.
> •    We have modified the statement of Theorem 3.1, 4.1 and the proof of Lemma 3.2 according to your suggestions.
>
> Regarding your question on how our insights could help practitioners in the future since we have characterized the convergence rate of gradient descent from the Gram matrix perspective, we believe our insights can inspire practitioners to design faster optimization algorithms from this perspective.
>
> We kindly ask you to read our revised paper and our response to common questions and re-evaluate your comments.
>
> We thank the reviewer again and welcome all further comments.

---

> > ### Comment · AnonReviewer4 · 2018-12-02
> > **I believe the revision addressed my concerns**
> >
> > The revised lemma is much clearer than the initial version.
> >
> > The proof of Lemma 3.2 only uses the randomness of w_i(0) s, and the result holds for any weight vectors satisfying the distance assumption in the Lemma, including the setting where the weight vectors are random and dependent on w_i(0) s.
> >
> > Typo: In the first line of Lemma 3.2, 'w_1, ..., w_m' should be 'w_1(0), ..., w_m(0)'.
> >
> > I have adjusted my score accordingly.

---

> > > ### Author Response · Authors · 2018-12-03
> > > **Thanks!**
> > >
> > > Thanks for increasing your score! We will fix the typo in our final version.

---

> > > ### Public Comment · (anonymous) · 2018-12-03
> > > **Random event independent of t but still depend on choice of w**
> > >
> > > Though assuming w fixed and only randomness of w(0), the random event still depends on w. I believe Lemma 3.2 actually proved that Prob[H(w) eigenvalues are lower bounded]>1 -delta, for any fixed w. But what is used in the latter proof seems to be Prob[for any fixed w, H(w) eigenvalues are lower bounded]>1 -delta.
> > >
> > > Think the following simple example.  If Z is N(0,1), then E[(Z-1)^2] = 2 and E[(Z+1)^2] = 2. By Markov inequality, Prob[(Z-1)^2 >2/delta]<delta, Prob[(Z+1)^2 >2/delta]<delta, but '(Z-1)^2 >2/delta' and '(Z+1)^2 >2/delta' are certainly different random events.

---

### Official Review · AnonReviewer3 · 2018-11-14

**Rating:** 3
**Confidence:** 5

**Review:**

Additional Review

This paper did NOT handle the non-differentiability and non-linearity very well. We can see this from the following three perspectives:

1. Proof idea: the proof of this paper is noisy version of the convergence analysis of  a simple convex problem --it treats the contribution of the non-linearity and non-differentiability as bounded noise.
2. The network size is of order n^6.
3. Network size requirement is dependent on \lambda_0.

1.Proof idea: The proof is essentially a noisy version of the convergence analysis of a linear regression problem provided in Appendix (at the end of this updated review). The only difference between linear regression and the problem in this paper is the changing patterns due to the non-linearity of ReLU. However, this paper views the changing patterns as noises compared to those unchanging patterns (e.g., S_i v.s. S_i^\perpendicular). The key trick is that if the actual trajectory radius (i.e.,the largest deviation from the initial point) R’ is much smaller than the desired trajectory radius R (given by a formula), then along the trajectory, the contribution of non-linearity is just O(n^2 R), which is small compared to the contribution of linearity, i.e., -\lambda_0 (shown in proof on page 9).

Following the above analysis, if the experiment shows that R’ is really small compared to R, then the approach of treating non-linearity as noise is fine. However, it is not the case for the problem studied in the experiments (Sec 5, Fig 1). In figure 1, we can easily see that the maximum distance R’ is O(1), which is far larger than R = c*\lambda_0/n^2 =10^-6 when n=1k. Therefore, the proof idea used in this paper is fundamentally not able to explain the phenomenon shown in the experiment. In fact, to address this issue, authors need to consider significant contribution of non-linearity, instead of just viewing them as noises.

2. The network size is too large. This paper requires O(n^6) neurons, that is 10^18 neurons for n=1000 samples used in the experiment. The theoretical trick to make R’< R is to note that R’ can be bounded by O(1/sqrt{m}) while R is independent of m, thus picking a sufficiently large m can make R’ very small. In a word, the reason that this paper requires so many neurons because of the inability of properly addressing non-linearity.

3. I found the dependence of the network size on the least eigenvalue funny, although the authors claim this tool is elegant. After authors add Thm 3.1 in the revision, I realize that the dependence on \lambda_0 might come from the fact that authors do NOT handle the issue of non-differentiability.

Let us see a simple example. Assume I have a dataset with \lambda_0 = 1. Now I am adding one more data point (x=0_d, y=1) to the dataset. After adding this sample, \lambda_0 clearly becomes 0. It seems I am just adding a constant 1 to the loss function and the gradient descent can also converge to the global min with a linear convergence rate since the constant does NOT contribution to the gradient. However, it seems the proof does NOT work. This is due to the fact that the “gradient” of the non-differentiable points are NOT well defined. Here is a simple example: h(w)=(y-ReLU(w*x))^2, where x= 0, y =1. By the definition provided in this paper (Eq.4), we can easily see that dh/dw = 1 for any w, even if h(w) = 1 for any w. This means that the constant can provide “fake” gradient information and make  the maximum distance become infinity, (R’=\inf). Therefore, the whole proof collapses. In fact, changing the gradient definition from I{z>=0} to I{z>0} does not address the issue and we can see this from this example w=g(w)=Relu(w)-Relu(-w) has a zero gradient at w=0.

In summary, the problem considered in this paper where the size m=O(n^6), maximum distance R’= O(1/n^2) is too easy compared to most problems in practice where m=\Theta(n), R’=O(1). To address the latter problem, we need a better definition of subgradient and need to analyze the significant contribution of non-linearity and non-differentiability, instead of just viewing them as noises.

=================================Appendix===============================

The proof basically follows from the convergence analysis of the following linear regression problem (note that u_j is fixed):
           \min_{w_1,...,w_m}\sum_{i=1}^{n}(f(x_i;w_1,...,w_m)-y_i)^2 = L(w_1,...,w_m)
where   f(x;w_1,...,w_m)=1/\sqrt{m}\sum_{j=1}^{m} a_j*(w_j^T x)*1{u_j^T x>=0}

Gradient Descent Algorithm:
-Initialization:
-For each j=1,...,m: a_j ~ U({-1,1}), u_j~N(0, I)
-Fix a_1,...,a_m, u_1,...,u_m
-Update:
-For t = 1,...,T
    w_j(t+1)  = w_j(t) - \eta* \nabla_{w_j}L(w_1,...,w_m) for j=1,..., m.

In this problem, since a_j and u_j are fixed, then model f is just a linear model w.r.t. w_j’s and the above problem is just a simple linear regression problem. Therefore, it is not difficult to prove the linear convergence rate for the gradient descent for the above problem under some mild assumptions.  Note that in this paper, u_j(t)= w_j(t) and are not fixed in iterations, i.e., patterns can change.

=========================
First, I apologize to the authors and ACs for the late review, since this paper desearves much more time to judge the quality.

Summary: This paper proves that the gradient descent/flow converges to the global optimum with zero training error under the settings (1) the neural network is a heavily over-parameterized ReLU network (i.e., requiting Omega(n^6) neurons); (2) the algorithm update rule “ignores” the non-differentiable point; (3) the parameters in the output layer (i.e., a_i’s) are fixed; (4) the data set has some non-degenerate properties and comes from a unit ball. The proof relies on the fact that the Gram matrix is always positive definite on the converging trajectory.

Pros: The proof is simple and seems to be correct. The paper is paper is written clearly  and easy to follow.

Cons:

The problem setting considered in this paper does not seem to be difficult enough. The difficulty of analyzing the landscape property of a ReLU network and proving the global convergence of the gradient descent mainly lies in the following three perspective and this paper does not try to tackle any one of them.

First, it is very hard to characterize the landscape or the convergence trajectory at/ near the non-differentiable point and this paper fails to touch it. The parameter space is separated into several regions by the hyperplanes and the loss function is differentiable in the interior of each region and non-differentiable on the boundary. I believe the very first question authors need to answer is wether there are critical points on the boundary and why the sub-gradient descent escapes from  any of these points. However, in this paper, authors avoid this problem by defining an update rule used  in practice and this rule does not use the sub-gradient at the non-differentiable point. Thus, it is totally unclear to me wether this global convergence result comes from the fact that this update rule can generally avoid the non-differentiable  points on the boundary or the fact that the landscape is so nice such that there are no critical points on the boundary or the fact that all points on the convergence trajectory is differentiable only in this unique problem.

Second, the problem is much easier if the loss is not jointly optimized over the parameters in the first and second layer. Having parameters in one layer fixed does not seem to be a big problem at first glance, but then I realize it indeed makes the problem much easier, which can be seen in the following example. If we randomly sample the weight vector w_i from N(0, I) and only optimize  over the parameters in the second layer, then it is straightforward to show the following result.

Result: If \lambda_\min(H^\inf)>0 and m=\Omega(n\log n), then with high probability, the loss function L is strongly convex with respect to a=(a_1,…, a_m) and the loss function is zero at the global minimum.

The above result shows that if we fix the parameters in the first layer and only optimize the parameters in the second layer, it is easy to prove the global convergence with a linear convergence rate. In fact, this result does not require the samples coming from a unit ball and the network size is only slightly over-parameterized. Therefore, if we are allowed to fix the parameters in some layer, how are the result presented in this paper fundamentally different from the above result.

Authors may say that the loss is not convex with respect to the weights in the first layer even if the second layer is fixed. However, when the second layer is fixed, the loss function is  smooth and convex in each parameter region and some recent works have shown that in this case, the loss function is a weakly global function. This means that the loss function is similar to a convex function except those plateaus and this further indicates that if the initial point is chosen in a strictly convex basin, the gradient descent is able to converge to a global min.  However, the problem becomes far more difficult if the loss is jointly optimized over  all parameters in the first and second layer. This can be easily seen since in each parameter region, the loss is no longer a convex function and this may lead to some high order saddle points such that the gradient descent cannot provably escape. Furthermore, the critical points on the boundary can be much more difficult to characterize for this joint optimization problem.


Third, the dataset considered in this paper does not seem to be a fundamental pattern and it seems more like a technical condition required by the proof. It is easy to see that a linearly separable dataset does not necessarily satisfy the conditions that 1) the gram matrix is positive definite and that 2) samples come from the surface of a unit ball. Therefore, I do not understand the reason why we need to analyze this pattern. Clearly, in practice, the data samples is unlikely sampled from a ball surface and it is totally unclear to me why the gram matrix is necessarily positive definite. I understand that some technical assumptions are needed in a theoretical work, but I would like to see more discussions on the dataset, e.g., some necessary conditions on the dataset such that the global convergence is possible.


Last, I understand that the over-parameterization assumption is needed. In fact, I expect the network size to be of the order Omega(n*ploylog(n)). I am wondering wether Omega(n^6) is a necessary condition or wether there exists a case such that Theta(n^6) is required.


Above all, I believe this paper is a half-baked paper with some interesting explorations. In summary, it cannot deal with non-differential points, which is considered a major difficulty for analyzing ReLU. In addition, it makes an un-justified assumption on some matrix, it requires too many neurons, and fixed 2nd layer. With so many strong assumptions, and compared to related works like [1], Mei et al., Bach and ..., its contribution is rather limited.

[1] https://arxiv.org/abs/1702.05777

---

> ### Author Response · Authors · 2018-11-19
> **Response to Reviewer 3**
>
> Thank for your long review. Unfortunately, we disagree with most of your comments. First, we would like to point out two wrong statements in your review.
>
> First, the “result” you claim is wrong. If the first layer is fixed and m = \Omega(n \logn), and only a=(a_1,…, a_m) is being optimized, this is a linear regression problem with respect to a=(a_1,…, a_m). Since m > n, this problem has more features than the number of samples, and the covariance matrix (Hessian) is degenerate. There is no way this problem is a strongly convex one.
>
> Second, you claimed there exists a linearly separable dataset whose corresponding H^{\infty} is degenerate. However, we are considering a regression problem whereas linearly separable condition is only a favorable condition for classification problems. We don’t understand what does linearly separable mean for regression.
>
> Now regarding your main complaint that the problem is not difficult enough:
>
> 1.    This is not true at all. Reviewer #1 and Reviewer #2 both explicitly agreed this is a challenging/difficult problem and we have devoted a whole paragraph (second paragraph on page 2) and many sentences in Section 2 to describe the difficulty.
>
>
> 2.    You complained that we are not analyzing the landscape of this non-differentiable function and we are using the “practically used update rule instead of subgradient.” We don’t understand the point here. Our primary goal is to understand why practically used rule (gradient descent) can achieve zero training loss. We have stated our goal at the beginning of the abstract and the introduction. For the non-differentiability issue, in the revised version we have cited papers and added discussions in the fourth paragraph of Section 2 on recent progress in dealing with non-differentiability.
>
>
> 3.    You claimed fixing one layer and optimizing the other one is a trivial problem. We agree if one fixes the first layer and optimizes the output layer, then this is trivial because this is a convex problem. However, if one fixes the output layer and optimizes the first layer, the problem is significantly harder. You claimed in this case
>
> “the loss function is a weakly global function. This means that the loss function is similar to a convex function except those plateaus and this further indicates that if the initial point is chosen in a strictly convex basin, the gradient descent is able to converge to a global min. ”
>
> We kindly ask for a reference and why it can imply the global convergence of gradient descent analyzed in our paper. To our knowledge, none of the previous results implies the global convergence of gradient descent in the setting we are analyzing. We have discussed this point in Section 2. Furthermore, we have never heard of the notion “weakly global function”.
>
> 4.    You believed that the inputs are generated from a unit sphere is a strong assumption. In our original version, we said making this assumption is only for simplicity. In our revised version, we added more details on this assumption. Please check footnote 7.
>
> 5.    For your other concerns, we kindly ask you to read our response to common questions.
>
>
> We thank the reviewer again. We welcome all further comments!

---

> ### Author Response · Authors · 2018-11-27
> **Response to Additional Review**
>
> Thanks for your thorough reading! We will add more discussions on non-differentiability!
>
> 1. Proof idea and experiments:
> We think viewing our proof from a "noisy" linear regression perspective is an interesting observation. Indeed, analyzing a hard non-linear problem from a "linear" perspective is a common practice in mathematics.
>
> In our proof, R' < R is a sufficient condition to show most patterns do not change, which we have verified in Figure 1 (b). It is possible that through other types of analysis, one can show most patterns do not change. For Figure 1 (c), we just want to verify that as $m$ becomes larger, the maximum distance becomes smaller.
>
> 2. Network size.
> We have discussed this point many times in the response.
> Our current bound requires m = \Omega(n^6). In this paper, to present the cleanest proof, we only use the simplest concentration inequalities (Hoeffding and Markov). We do not think this bound is tight, and we believe using more advanced techniques from probability theory, this bound can be tightened.
>
> 3. Dependency on lambda_0.
> First of all, your example is not valid in our setting. If x=0, y=1, it is not possible that ReLU(w*x) can achieve zero training error.
> Furthermore, it is easy to prove linear convergence for your example because we can just study the Gram matrix defined over other data points, which has a positive lambda_0.
> We will add a remark about this in our final version. Thanks for pointing out.

---

### Public Comment · (anonymous) · 2018-10-05
**Lack of Experimental Results and Unrealistic Assumptions?**

The analysis in this paper seems technically sound. However, I have questions w.r.t. this paper: is there any experimental result to support the analysis in this paper? The results are quite simple, and I wish the author(s) could add some experimental validations, even a toy one, to support the theoretical results.

Besides, the assumption on the least eigenvalue of the Gram matrix seems somewhat unreasonable, because if we use some data augmentation tricks, such as mix-up [1] (i.e. if there is a training sample that is the linear combination of other samples), the assumption apparently does not hold in this case in the sense that the least eigenvalue of this gram matrix will become zero. However, the adding of one more data seems have little influence on the training procedure.

Another concern is that the analysis and conclusion in this paper is somewhat trivial. There are not much technical contributions in this paper. The technical part follows closely to this work [2].


[1] Zhang, Hongyi, et al. "mixup: Beyond Empirical Risk Minimization." (2018).
[2] Li, Yuanzhi, and Yingyu Liang. "Learning Overparameterized Neural Networks via Stochastic Gradient Descent on Structured Data." arXiv preprint arXiv:1808.01204 (2018).

---

> ### Author Response · Authors · 2018-10-06
> **The Gram matrix is not degenerate. The analysis is simple and novel.**
>
> We thank for your comments and we are happy to answer your concerns.
>
> 1) Adding a linear combination of existing features to the data set leads to a degenerate Gram matrix?
> This is wrong. Every entry in our Gram matrix is not an inner product between two features, but the result of using a non-linear kernel acting on two features. Please check our definition of the Gram matrix (H^{\infty}) more carefully (c.f. Theorem 3).
>
> For data augmentation with a linear combination of other samples, here we provide a counterexample.
> We have two features (1,0), (0,1) and we add a linear combination (1/\sqrt{2},1/\sqrt{2}).
> The Gram matrix is
> [0.5000         0    0.2652;
> 0    0.5000    0.2652;
> 0.2652    0.2652    0.5000 ]
> which is not degenerate.
> In general, only if the activation is linear, the Gram matrix becomes degenerate after adding a linear combination of existing features.
>
> In fact, we can easily prove as long as no two features are parallel, H^\infty is always non-degenerate. We will add the proof in the revised version.
>
> 2) Is this a trivial paper?
> Simplicity is not equivalent to triviality.
>
> Our result is simple: we just prove randomly initialized gradient descent achieves zero training loss for over-parameterized neural networks with a linear convergence rate. However, why randomly initialized first order methods can fit all training data is one of the unsolved open problems in neural network research.
>
> For the same setting (training two-layer ReLU activated neural networks), there are many previous attempts to answer this question but these results often rely upon strong assumptions on the labels and input distributions or do not imply why randomly initialized first order method can achieve zero training loss. Please see the second paragraph on Page 2 and Section 3 for detailed discussions.
>
> For technical contributions, we do agree our analysis is simple but we think this is actually an advantage because it will be easier to generalize simple arguments instead of involved ones. Our proof does not require heavy calculations and reveals the intrinsic properties of over-parameterized neural networks and random initialization schemes. Please see Analysis Technique Overview paragraph on page 2.
>
> Comparing with [2], except that we use the same property that the patterns do not change by much during training, our analysis is completely different from theirs and is significantly simpler and more transparent. We have devoted a whole paragraph in Section 3 discussing the differences with [2].
>
> 3) Experiments
> We would like to emphasize that this is pure theory paper and the theorem we proved (randomly initialized gradient descent achieves zero training loss) is a well known experimental fact in training neural networks. Nevertheless, we are happy to provide some experimental results in the revised version.

---

> > ### Public Comment · (anonymous) · 2018-10-07
> > **Thanks for Your Reply**
> >
> > Thanks for the author(s) reply.
> >
> > I've just seen some discussions about this paper on another website and here I wanna seen the official reply from the author(s) w.r.t. the following interesting comments, which is also somewhat the concerns of mine. (I simplily repost those discussions)
> >
> > 1. "One of the mystery in the success of neural networks is randomly initialized first order methods like gradient descent can achieve zero training loss even though the objective function is non-convex and non-smooth. This paper demystifies this surprising phenomenon for two-layer fully connected ReLU activated neural networks."
> > Hardly a mystery, Cybenko's paper back in 1989 pointed out NN with one hidden layer can approximate any continuous high-dimensional surface without higher degree of smoothness assumption nor being convex, optimization methods like gradient descent is but one of the methods can do the job.
> >
> > 2."For an m hidden node shallow neural network with ReLU activation and n training data, we show as long as m is large enough and the data is non-degenerate, randomly initialized gradient descent converges a globally optimal solution with a linear convergence rate for the quadratic loss function."
> > Another falsehood, the assumption of surface with positive eigen-values i.e. non-degenerate (in theorem 3.1 and 4.1 for example) implies convexity of the solution landscape. When the data is non-convex, there is no guarantee nor proof that the gradient descent or other more powerful optimization methods can always find the global optimal. Non-convexity problems pose similar challenges like NP-hard problems: solutions stuck in local optimum and there is no way in general to convert locally best solution to global optimal.
> >
> > 3."Cybenko's paper back in 1989 pointed out NN with one hidden layer can approximate any continuous high-dimensional surface without higher degree of smoothness assumption nor being convex."
> > Cybenko's paper only says that, for a given continuous function and epsilon, there exists a one-hidden-layer sigmoidal NN with less than epsilon maximum error. It says nothing about the learnability of this NN (nor even the number of neurons in it).
> >
> > 4. "the assumption of surface with positive eigen-values i.e. non-degenerate (in theorem 3.1 and 4.1 for example) implies convexity of the solution landscape."
> > The matrix H∞ is not the "solution landscape". It's a function of the data only, not the parameters. It is not the Hessian of the loss function, as you seem to think.
> >
> >
> > 5."The key assumption is the least eigenvalue of the matrix H∞ is strictly positive. Interestingly, various properties of this H∞ matrix has been thoroughly studied in previous work [Xie et al., 2017, Tsuchida et al., 2017]. In general, unless the data is degenerate, the smallest eigenvalue of H∞ is strictly positive."
> > For example, Xie's paper [1] focus most with spherical data, from section 3. Problem setting and preliminaries.
> >
> > 6."We will focus on a special class of data distributions where the input x ∈ Rd is drawn uniformly from the unit sphere, and assume that |y| ≤ Y . We consider the following hypothesis class."
> > Moreover, it also stated:
> > "Typically, gradient descent over L(f) is used to learn all the parameters in f, and a solution with small gradient is returned at the end. However, adjusting the bases {wk} leads to a non-convex optimization problem, and there is no theoretical guarantee that gradient descent can find global optima."
> >
> > It said nothing how common or not a given data set is convex like the current paper claimed. We suspect not, in general. Xie mentioned nothing about such data being degenerate.
> >
> > Now to Cybenko's paper:
> > "Cybenko's paper only says that, for a given continuous function and epsilon, there exists a one-hidden-layer sigmoidal NN with less than epsilon maximum error. It says nothing about the learnability of this NN (nor even the number of neurons in it)."
> >
> > Once we know the objective function, and expression of functional form, then the number of hidden layer neurons is a matter of engineering as long as we know "there exists a one-hidden-layer sigmoidal NN with less than epsilon maximum error.", that's learnability of one hidden-layer NN.
> >
> > References:
> > [1]. Xie, Bo, Yingyu Liang, and Le Song. "Diverse neural network learns true target functions." arXiv preprint arXiv:1611.03131 (2016).

---

> > > ### Author Response · Authors · 2018-10-10
> > > **Reply**
> > >
> > > Thanks for bringing concerns from others! We are happy to answer these concerns. In fact, point 3 and point 4 already resolved some of the issues.
> > >
> > >
> > > To point 1: Comparison with the universal approximation theorem.
> > > Response: The universal approximation theorem only establishes that there exists a wide neural network that can approximate continuous function on compact subsets of $R^d$. It does not imply a wide neural network trained by randomly initialized gradient descent has the same approximation property.
> > > We will add more discussions on universal approximation theorem in the revised version.
> > >
> > > To point 2: Is this a convex problem?
> > > Response: because of the use of ReLU activation, this is not a convex problem. The l2 loss is convex with respect to the predictions but is not convex with respect to the parameters we are optimizing.
> > >
> > > To point 5,6: Degeneracy of the Gram matrix:
> > > Response: This has been addressed in our previous reply.

---

### Public Comment · ~Hongyi_Zhang1 · 2018-10-07
**possible simplification of H^{\infty}**

Interesting results! It seems to me that the definition of H^{\infty}_{ij} in your main theorems could be simplified as (x_i^T x_j) * arccos(- x_i^T x_j) / (2 * pi) -- am I correct?

---

> ### Author Response · Authors · 2018-10-10
> **Correct!**
>
> Yes that's the correct formula. Thanks!

---

### Public Comment · (anonymous) · 2018-10-07
**comments on relation with prior works**

I would like to give a comment on the relation of this paper and certain prior works. The paper by Chizat and Bach proves continuous-time gradient flow can converge to optimal population loss, in the limit of infinite number of neurons, under certain conditions (which include sigmoid activation, and ReLU at a formal level). Mei et al. proves that noisy SGD can optimize to near optimal population loss. In fact, Mei et al. provides a quantitative statement, that the continuous-time flow and the discrete-time one are close already when the number of neurons >> the dimension of the input (i.e. m>>d as in the notation of this paper). As such, these works already suggest that first-order methods can work well on neural nets with a single hidden layer (in terms of population loss), requiring m>>d.

These two works are briefly mentioned in the paper, but I think it is important to clarify the distinction. The paper, whose analytical approach aligns with many other papers, proves that gradient descent can optimize to optimal empirical loss, for the specific case of ReLU activation. The analysis is nice in its simplicity (and length!), and so I believe many will try to study  this type of analysis. The key finding is that when m>>poly(n) (where n is the number of training samples) and when n is large, many things remain close to initialization at all iterations. As such, random initialization works to our advantage.

Interestingly the aforementioned two works require m>>d, whereas here m>>poly(n). There is no contradiction since the former analyzes SGD, and this paper analyzes (full-batch) gradient descent. Yet this difference raises a question of whether there is an analysis to unify the picture. There is also a question of generalization performance, which is resolved in the aforementioned two works but not in this paper.

I must admit that I have not verified the proof, so it remains to see whether the analysis is correct.

As a clarifying question, is it crucial that the output weight is initialized uniformly random? The role of random initialization for the output weight is not transparent at first glance.

---

> ### Public Comment · ~Danica_J._Sutherland1 · 2018-10-08
> **Comment on output weights**
>
> Not an author and haven't super-carefully checked the proof, but the derivation of (5), at the start of Proof of Theorem 3.1, assumes that a_r^2 = 1. Otherwise H would contain an a_r^2 term multiplying the indicator; if you used a different distribution for a, then everything to do with H is going to depend on that too. That could make things a lot messier....
>
> But that doesn't prevent you from choosing a_r as some weighted distribution on +-1. In particular, you could pick all of the a_r = 1. The only place I see that affecting the continuous-time proof is the Markov's inequality bound for ||y - u(0)|| at the end, which uses E[a_r] = 0. But if you had some other high-probability bound on ||y - u(0)||, which you could definitely get just based on the distribution of W, it seems that the rest of the proof carries through with possibly a bigger m. But that can't be right – if all the a_r = 1, f can't output negative values, and nothing else stops any of the y from being negative.... Authors, what am I missing here?

---

> > ### Public Comment · (anonymous) · 2018-10-08
> > **Reply**
> >
> > Thank you Dougal. The assumption on a_r is stated in Theorem 3.1, that a_r \in {-1, 1} (and hence a_r^2=1). This is a perfectly fine assumption for ReLU given its homogeneity. There is also randomization: a_r ~ Unif({-1, 1}). Somehow the role of randomness of a_r is not transparent in the proof. But I suspect it should be important: suppose that I use a_r = 1 for all r (hence no randomization), and since ReLU is non-negative, with strictly negative labels y, there is no way that the network can find y...
> >
> > P.S.: I somehow missed the second part of Dougal's reply, which pointed to the same concern.

---

> ### Author Response · Authors · 2018-10-11
> **Thanks for your comments and questions!**
>
> Thanks for your comments and questions!
>
> As stated in our paper, the results in the two papers you mentioned do not imply why randomly initialized gradient descent can achieve 0 training loss with arbitrary labels. Furthermore, there are many subtle differences in the assumptions. We will definitely expand our discussions on these two papers in the revised version.
>
>
> Dependency on d and n: our bound depends on lambda_0, which is a dataset-dependent quantity. In general, this quantity is related to d,n and the input distribution.
>
>
> On generalization: in general, population risk bound can be obtained only if there are additional assumptions on the input distribution and labels. It is an interesting direction to extend our analysis to incorporate structures in the input distributions and labels.
>
>
> Why using uniform random initialization for the second layer:
> There are two purposes for using this initialization scheme.
> First, as already explained by Dougal, $a_r^2 =1$ makes H matrix independent of a_r and in turn, makes our calculation much easier.
> Second, this initialization makes ||y-u(0)||_2 = O(\sqrt{n}). If the output layer are all ones, then u(0) is of order \sqrt{m}  which makes  ||y-u(0)||_2  be of order \sqrt{mn}. In this case, R' cannot be smaller than R.

---

### Public Comment · ~Danica_J._Sutherland1 · 2018-10-08
**some thoughts + a bit of a numerical study of lambda_0**

We discussed this in our reading group today, and I'd like to relay some of our thoughts to other readers.

The paper randomly initializes an extremely overparameterized network: m = Omega(n^6 / lambda_0^4), where lambda_0's dependence on n will vary with the dataset, but presumably it decays with n, making the overall rate for m worse than n^6. Then, here's another way to think about the results of the paper; with high probability:

1. There is a global optimum without switching any of the activation patterns, i.e. keeping sign(w_r^T x_i) the same for all i, r. (This isn't directly shown as a separate step in the paper, but it's implied by Theorem 3.1.)

2. Following a continuous-time gradient flow leads you to that global optimum, following a path that "looks" strongly convex as you follow it (so you get linear convergence), without ever switching any of the sign(w_r^T x_i), with high probability.

3. Discrete-time gradient descent, for a small enough step size O(lambda_0 / n^2), does basically the same thing. It's allowed to switch some of the activation patterns, but only a few of them, S_i (or maybe S_i^\perp, depending on if you go by the definition you give or the way you then use it...). Those ones don't affect the loss too much, and we still have convergence.


Given (1), (2) is maybe not super-surprising: the set of W with the same activation patterns is the intersection of m n linear constraints, and within that set, the objective function is a convex QP. Probably lambda_min(H(0)) is related to lambda_min of the quadratic term in the QP objective, though I couldn't immediately show that. Of course, this doesn't show a result as strong as (2)/(3) without additionally showing you don't happen to break the constraints in following the gradient flow, and and it's circular anyway in that it's not obvious how to show (1) other than through the proof via (2) given here.


The applicability of this approach to deeper networks, then, rests on how realistic the extreme overparameterization here is. Is it still the case that you can avoid switching too many activation patterns in training a deeper network? It would be interesting to track that empirically while training a practical deep net. If switching activation patterns is indeed rare, then this type of approach might be very fruitful for studying deeper nets. Even if not, though, this is an elegant solution to the 1-layer setting.


Out of curiosity, I also tried to check numerically what the dependence of lambda_0 is on n for a uniform distribution of inputs. It seems like lambda_0 is about n^{-2} for d=2, n^{-1/2} for d = 5, and n^{-1/4} for d = 10 - https://gist.github.com/dougalsutherland/cc7d8b6d740c6c07d3c6081cfb42d191 . If that's correct, then in 2d the required m is Omega(n^14) (!) while in 10d it's only Omega(n^7), and presumably in very high dimensions it becomes omega(n^6). It might be interesting to try to actually bound lambda_0 in terms of n and d to see if these simulations are accurate. (It might very well be that lambda_0 has a different rate for very large n, with "very large" depending on d; I only ran up to n about 3,000 because I only wanted to run for a few minutes on my desktop.)

---

> ### Author Response · Authors · 2018-10-11
> **Thanks for your comments and the numerical study!**
>
> Thanks for your comments and the numerical study! They are very inspiring!
>
> For the analysis:
> Your intuition is basically correct. We want to clarify that our current proof cannot show for continuous time gradient flow there is no activation pattern change. What we can show is the number of pattern changes is small and only incur small perturbation on H. See Lemma 3.2 and its proof.
>
> Extension to deep neural networks:
> Yes, it would be very interesting to investigate empirically whether there is only a small amount of pattern changes when training deeper models.
>
> On lambda_0:
> Thanks for your numerical study! We agree it would be very interesting to obtain some bounds on lambda_0 under certain distributional assumptions.

---

> > ### Public Comment · ~Danica_J._Sutherland1 · 2018-10-11
> > **Change in activation patterns**
> >
> > I see now; Lemma 3.2 says that the expected number of total changes is small, not zero. Whoops; thanks.

---

> ### Public Comment · ~Olivier_Grisel1 · 2018-10-29
> **Another numerical study**
>
> Interesting numerical study. I did not know about the analytical relationship between H and the data Gram matrix. I did a more brute-force numerical study of H on a non-random toy dataset (8x8-pixels gray level digits, d=64, n~=1797) and found lambda_0 > 1.3e-2 which is in line with your random data study:
>
> https://gist.github.com/ogrisel/1b430b2bf1e83173f6061676c62b9f18

---

> > ### Author Response · Authors · 2018-11-19
> > **Thanks for your experiments**
> >
> > Thanks, Olivier.
> >
> > We have acknowledged your experiments in our response.

---

### Public Comment · (anonymous) · 2018-10-26
**Lemma 3.2 and w.h.p. in proving induction hypothesis**

Thanks for the inspiring work. I found something confusing about the probability part though.

Denote B(t) = the event that at time/iteration t, || w_r(t)-w_r(0)||_2\leq R for all r happens. Denote C(t) =  the event that at time/iteration t, the smallest eigenvalue of H(t) is at least \lambda_0/2 happens

Then Lemma 3.2 states that the conditional probability Prob[C(k)|B(k)] is large( > 1- c) when c ~ R*n^2/\lambda_0 for a fixed k. However, it is unclear whether C(k)|B(k) implies C(k+1)|B(k+1) from the paper. It is possible that Prob[\cap_k=1^N (C(k)|B(k))] is not high at all, and even could be zero when N approaches infinity.

In the last few lines of proving the induction hypothesis on page 10,  it uses Lemma 3.2 that C(k)|B(k) holds with high probability over initialization. But if we review the WHOLE process of proof by induction, in k=1,2.. till infinity, we assume different events hold (assume C(k)|B(k) when proving case k+1), and their relationships are unclear. Thus the "with high probability" statement seems to be not solid to me. No lower bound on Prob[\cap_k=1^\infty (C(k)|B(k))] is proved.

I would really appreciate your answer to this!

---

> ### Author Response · Authors · 2018-10-27
> **Clarification**
>
> Thanks for your question.
>
> We proved that with high probability over initialization, for any weight matrix $W(t)$ that satisfies $w_r(t)$ is close to $w_r(0)$ for all $r \in [m]$,  the induced Gram matrix $H(t)$ has lower bounded eigenvalue. Here $t$ is just an index relating the weight matrix and the induced Gram matrix. Note there is only one event which is independent of $t$.
>
> We are sorry about the confusion and we will modify the statement of the lemma to make it more clear in the revised version.

---

> > ### Public Comment · (anonymous) · 2018-10-29
> > **Reply**
> >
> > Thanks for your reply, but sorry, I couldn't see how it helps to answer my question. Looking forward to the revision though.

---

### Public Comment · ~Olivier_Grisel1 · 2018-10-29
**non-degenerate data**

I found the paper interesting to read (although I did not try to check the mathematical correctness of the results).

One point could be improved though: several times the text mentions that the main assumption is that "data is non-degenerate" without formally defining what is meant by this. The data matrix is not square so the traditional definition of non-degeneracy does not apply here.

When reading the theorems, I believe that the informal "non-degenerate data" assumption of the main text corresponds to the double assumptions that each input vectors has unit norm and more importantly that the H_inf kernel matrix is full-rank (non-degenerate).

In practice, this full-rank H_inf kernel assumption is typically not met if there exists duplicated samples in the training set (if there are duplicated samples with different labels, it's not possible to have zero training loss for any model).

I just read in your reply (https://openreview.net/forum?id=S1eK3i09YQ&noteId=SJeNOPOU9Q) that you can prove that this assumption is met as soon as there are no two parallel samples in the training set. But I assume that this is not necessarily a problem if the labels of such parallel samples are the same. Furthermore, since you also assume that all x_i have unit norm, a pair of parallel samples is actually a pair of duplicated samples.

So to conclude I would suggest editing your text to change the "non-degenerate data" phrase to something more specific (such as "record-wise normed data without duplicated records" or alternatively "non-degenerate extended feature matrix") so as to avoid any confusion.

---

> ### Author Response · Authors · 2018-11-19
> **Thanks for your suggestion!**
>
> We thank for your suggestion. We have changed "non-degenerate data" to "no parallel inputs".

---

### Author Response · Authors · 2018-11-19
**Response to Common Questions and Summary of Revisions**

Dear reviewers,

We thank for all your comments. Especially all reviewers agree our proof is simple. Here we address some common questions from reviews and other comments.

1.    H^{\infty} matrix.
Many comments asked when the least eigenvalue of H^{\infty} is strictly positive and what is the intuition of H^{\infty} matrix. We thank Dougal J Sutherland and Olivier Grisel for providing numerical evidence showing that on real datasets, this quantity is indeed strictly positive.

a.    Theoretically, in our revised version, we give a theorem (c.f. Theorem 3.1) which shows if no two inputs are parallel, then the H^{\infty} is full rank and thus it has a strictly positive eigenvalue.

b.    Here we also want to discuss informally on why we think H^{\infty} is the fundamental quantity that determines the convergence rate. In Equation (7), the time derivative of the predictions u(t) is EQUAL to -H(t) (y-u(t)), i.e., the dynamics of the predictions is completely determined by H(t). Furthermore, in our analysis, we show if m -> \infty, H(t) -> H^{\infty} for all t >0. Therefore, the worst case scenario is that at the beginning y-u(0) is in the span of the eigenvector of H^{\infty} that corresponds to the least eigenvalue of H^{\infty}. In this case, y-u(t) will stay in this space and by one-dimensional linear ODE theory, we see that y-u(t) converges to 0 at a rate exp(-\lambda_0 t). Also, see Remark 3.1.



2.    Why fixing the output layer and only training the first layer? The analysis will be much harder if one trains both the first and the output layer.
This is the concern raised by Reviewer 2 and Reviewer 3. In our original version, we only analyzed the convergence of gradient optimizing the first layer because we believe this problem already demonstrated the main challenge as many previous works tried to understand the same problem, but none of them has a polynomial time convergence guarantee towards zero training loss. For reviewers’ concern:

a.    First, we disagree with Reviewer 3 that analyzing the case that only the first layer is trained is a trivial problem. For the same setting, there are many previous attempts to answer this question, but these results often rely upon strong assumptions on the labels and input distributions or do not imply why randomly initialized first order method can achieve zero training loss. Please see the second paragraph on Page 2 and Section 3 for detailed discussions.

b.    Second, if we fix the first layer and only train the second layer, the learned function is different from the function learned by fixing the second layer and training the first layer. We have added this point in footnote 3.

c.    Lastly, in our revised version, we added a new theorem (c.f. Theorem 3.3) which shows using gradient flow to train both layers jointly, we can still enjoy linear convergence rate towards zero loss. To prove Theorem 3.3, we use the same arguments as we used to prove Theorem 3.1 with slightly more calculations. Therefore, we have shown analyzing the case that both layers are trained is just as hard as analyzing the case where only the first layer is trained.


3.    Amount of over-parameterization.
Our current bound requires m = \Omega(n^6). In this paper, to present the cleanest proof, we only use the simplest concentration inequalities (Hoeffding and Markov). As we discussed in the conclusion section, we do not think this bound is tight, and we believe using more advanced techniques from probability theory, this bound can be tightened.



4.    Lemma 3.2.
We are sorry about the confusion in the statement in our original version. We have changed the statement, and the new statement is independent of t.



5.    Extending to more layers.
We have added more discussions in the conclusion section on how to extend our analysis to deeper neural networks, including a very concrete plan. In short, for deep neural networks, we can also consider the dynamics of the n predictions, and the dynamics are determined by the summation of H (number of layers) Gram matrices. We conjecture that 1) at the initialization phase as m -> \infty, the summation converges to a fixed n by n matrix and 2) as m -> \infty, these matrices do not change by much over iterations. Thus, as long as the least eigenvalue of that fixed matrix is strictly positive and m is large enough, we can still have linear convergence for deep neural networks.




Summary of Revisions:
1.    We add a new theorem (Theorem 3.1) which shows as long as no two inputs are parallel, H^{\infty} matrix is non-degenerate.
2.    We add a new theorem (Theorem 3.3) on the convergence of gradient flow for jointly training both layers.
3.    We add experimental results to verify our theoretical findings.
4.    More discussions on how to extend our analysis to more layers and why H^{\infty} is a fundamental quantity.

---

### Public Comment · (anonymous) · 2018-11-23
**The impact of the current paper looks very limited**

Although this paper provides a theoretical guarantee for one hidden layer ReLU based neural networks, the proposed analysis seems very limited, and I’m wondering whether this analysis can give us some insights for analyzing deep networks to get meaningful results.

In detail, the lower bound assumption of H will introduce a quantity \lambda_0 into the dependence of m. This quantity can be extremely small in the case of deep networks, which gives us meaningless requirement of the number of hidden nodes. Most part of the current paper discuss about the continuous time analysis. However, this kind of analysis can get rid of the smoothness requirement of the loss function, which is one of the biggest challenges for analyzing ReLU based networks. In addition, the discrete time analysis is based on some loss concentration bounds, which may lead to meaningless results for deep networks.

I think the proposed analysis of the current paper looks very limited.

---

> ### Author Response · Authors · 2018-11-26
> **Response**
>
> Thanks for your comments. However, we disagree with your comments.
>
> First, this paper is only about the training error, so we are confused why you talked about sample complexity.
>
> Second, you wrote, “\lambda_0 can be extremely small in the case of deep networks”. However, you did not give any concrete evidence about this claim.
>
> Third, we are confused about why using continuous analysis to gain intuition is a wrong approach. Many previous papers used this approach to analyze convex optimization problems and deep learning optimization problems [1,2,3,4].
>
> Fourth, you wrote “the discrete analysis based on some loss concentration bounds, which may lead to meaningless results for deep networks.” Again, you did not give any concrete evidence on why for deep networks our analysis will be meaningless, and we are confused about what are the “loss concentration bounds” you are referring to.
>
> [1] Ashia C Wilson, Benjamin Recht, and Michael I Jordan. A Lyapunov analysis of momentum methods in optimization. arXiv preprint arXiv:1611.02635, 2016.
> [2] Zhang, J., Mokhtari, A., Sra, S., & Jadbabaie, A. (2018). Direct Runge-Kutta discretization achieves acceleration. arXiv preprint arXiv:1805.00521.
> [3] S Sanjeev Arora, Nadav Cohen, and Elad Hazan. On the optimization of deep networks: Implicit acceleration by overparameterization. arXiv preprint arXiv:1802.06509, 201
> [4] Simon S Du, Wei Hu, and Jason D Lee. Algorithmic regularization in learning deep homogeneous models: Layers are automatically balanced. arXiv preprint arXiv:1806.00900, 2018.

---

> > ### Public Comment · (anonymous) · 2018-11-27
> > **Further discussion**
> >
> > Thanks so much for your response. I would like to clarify my concerns.
> >
> > 1. I mean the number of hidden nodes m will depend on \lambda_0.
> >
> > 2. The current paper fails to give an explicit relationship between \lambda_0 and n, thus the requirement of m may be meaningless. What if this dependence is exponential? The authors should at least prove that the dependence of \lambda_0 on n is polynomial under some more natural assumptions on data distribution. If this cannot be proved, does it imply that this eigenvalue lower bound assumption hides the major difficulty of this problem?
> >
> > 3. In the current paper, the authors provide (i) continuous time convergence result (ii) discrete time convergence result (iii) discussion on how the proof method for continuous case can be generalized to deep networks. However, the connection between the continuous time analysis and the discrete time analysis is unclear in the current paper. It seems that the current discrete time analysis is not really a discretization of the continuous time proof, and the proof method looks independent of the continuous time analysis. As a result, it is unclear if the current discrete time analysis can provide enough insight on the training of deep networks, especially since the non-smoothness of ReLU activation function is one of the major difficulties.

---

> > > ### Author Response · Authors · 2018-11-27
> > > **Thanks for your comments!**
> > >
> > > Thanks for your clarifications and we are happy to address your concerns.
> > >
> > > 1.    On H^{\infty} and \lambda_0.
> > > We have discussed this in length in our Response to Common Questions and Summary of Revisions. In short, because Equation (7) is an equality, at least in the large $m$ regime, $H^{\infty}$ determines the whole optimization dynamics, and as a consequence, $\lambda_0$ is the correct complexity measure. See more discussions in Remark 3.1.
> > >
> > > We are not hiding the difficulty because we have identified the correct complexity measure. We believe it is indeed an interesting problem about how the spectrum of $H^{\infty}$ is related to other assumptions on the training data. We will list this problem in the Discussion section in our final version.
> > >
> > > As a side note, before this paper, even if we allow $m$ to be exponential in $n$, there is no analysis showing that randomly initialized gradient descent can achieve zero training loss.
> > >
> > > 2.    On discrete time analysis.
> > > Our discrete time analysis follows closely to the continuous time analysis. Note we analyze $u(k+1) – u(k)$ which is analog to $du/dt$. Furthermore, in the equation in the middle of page 9, in the third equality, we decompose the loss at the (k+1)-th iteration into several terms. Note the second term just corresponds to $d(\|y-u(t)\|_2^2)/dt$ in the proof of Lemma 3.3 and the other terms are perturbations terms due to discretization. We will make the connection between continuous time analysis and discrete time analysis clearer in our final version. Thanks for pointing out!

---

> ### Public Comment · (anonymous) · 2018-12-02
> **One of the best papers in ICLR**
>
> This paper seems to be one of the most popular papers in ICLR though. It got a lot of attention on social media as well as in academia. The impact of the paper is definitely huge as it's closely correlated with the popularity.

---

### Public Comment · (anonymous) · 2018-11-28
**Interesting paper and recent follow-ups**

This paper seems to be an interesting and important paper for neural networks theory. It gets rid of the distributional input assumption common in previous works. It also gives a linear convergence rate which could not be made possible solely by landscape analysis.

In the analysis of this paper, the H^infty matrix appears naturally and seems to reveal a connection between neural networks and kernels. Moreover, I would like to mention that the ideas presented in the current submission have recently been generalized to deal with multi-layer neural networks, which clearly illustrates the potential its proof structure and techniques.

---

> ### Author Response · Authors · 2018-11-28
> **Thanks!**
>
> Thanks for your encouraging comments!

---

### Public Comment · (anonymous) · 2019-07-25
**Proof of Lemma 3.1**

Thank authors for the interesting work. We found some parts of proof of Lemma 3.1 (page 15) to be confusing. Any clarification is greatly appreciated!

(a) Could anybody please elaborate on how to derive 2nd inequality formula from 1st inequality formula in Proof of Lemma 3.1? It writes "Setting δ′ = n^2 δ and applying union bound over (i,j) pairs", but how does the upper bound
    2 \sqrt{log(1 / δ')} / \sqrt{m}
relax to
    4 \sqrt{log(n / δ)} / \sqrt{m}
by setting "δ' = n^2"?

(b) Besides, we also find it confusing to derive the 1st inequality by Hoeffding's inequality. The paper writes: with probability 1 - δ', we have
    |H_{ij}(0) - H_{ij}^∞| ≤ 2 \sqrt{log (1 / δ')} / \sqrt{m}.

But using the Hoeffding's inequality (as formulated in Corollary 7 of lecture note [1]), it derives an upper bound to be
    \sqrt{log(2 / δ')} / \sqrt{2m}
instead of the
    2 \sqrt{log (1 / δ')} / \sqrt{m}.

Are we wrong anywhere in understanding the proof? Thanks a lot in advance.


[1] http://www.stat.cmu.edu/~larry/=stat705/Lecture2.pdf

---

### Meta-Review · Area_Chair1 · 2018-12-12
**ICLR 2019 decision**

**Confidence:** 4
**Recommendation:** Accept (Poster)

**Metareview:**

This paper proves that gradient descent with random initialization converges to global minima for a squared loss penalty over a two layer ReLU network and arbitrarily labeled data. The paper has several weakness such as, 1) assuming top layer is fixed, 2) large number of hidden units 'm', 3) analysis is for squared loss. Despite these weaknesses the paper makes a novel contribution to a relatively challenging problem, and is able to show convergence results without strong assumptions on the input data or the model. Reviewers find the results mostly interesting and have some concerns about the \lambda_0 requirement. I believe the authors have sufficiently addressed this issue in their response and  I suggest acceptance.